# The synergistic effect of Hf-O-Ru bonds and oxygen vacancies in Ru/HfO$_2$ for enhanced hydrogen evolution

Guangkai Li[1,5], Haeseong Jang [2,5], Shangguo Liu[1,5], Zijian Li[3], Min Gyu Kim [4], Qing Qin [1✉], Xien Liu [1✉] & Jaephil Cho [2✉]

Ru nanoparticles have been demonstrated to be highly active electrocatalysts for the hydrogen evolution reaction (HER). At present, most of Ru nanoparticles-based HER electrocatalysts with high activity are supported by heteroatom-doped carbon substrates. Few metal oxides with large band gap (more than 5 eV) as the substrates of Ru nanoparticles are employed for the HER. By using large band gap metal oxides substrates, we can distinguish the contribution of Ru nanoparticles from the substrates. Here, a highly efficient Ru/HfO$_2$ composite is developed by tuning numbers of Ru-O-Hf bonds and oxygen vacancies, resulting in a 20-fold enhancement in mass activity over commercial Pt/C in an alkaline medium. Density functional theory (DFT) calculations reveal that strong metal-support interaction via Ru-O-Hf bonds and the oxygen vacancies in the supported Ru samples synergistically lower the energy barrier for water dissociation to improve catalytic activities.

[1] College of Chemical Engineering, Qingdao University of Science and Technology, Qingdao, China. [2] Department of Energy Engineering, Department of Energy and Chemical Engineering, Ulsan National Institute of Science and Technology (UNIST), Ulsan, South Korea. [3] Department of Chemistry, City University of Hong Kong, Hong Kong, China. [4] Beamline Research Division, Pohang Accelerator Laboratory (PAL), Pohang, South Korea. [5]These authors contributed equally: Guangkai Li, Haeseong Jang and Shangguo Liu. ✉email: qinqing@qust.edu.cn; liuxien@qust.edu.cn; jpcho@unist.ac.kr

Hydrogen produced from water splitting powered by various renewable energy sources is regarded as a sustainable and clean energy alternative to non-renewable, reserveless, and environmentally unfriendly fossil fuels[1–5]. The alkaline-water electrolysis technology has been commercialized, in which Ni or Fe mesh is generally used as the electrocatalyst. The current density and energy efficiency of this technology are ~0.25 A/cm$^2$ and 60%, respectively[6], which can be further improved by developing more highly active electrocatalysts. Among the known electrocatalysts, platinum group metals and alloys show excellent activities for the hydrogen evolution reaction (HER), for example, Pt/C is the benchmark electrocatalyst for HER, and its analogs Ru/C also exhibits a large room for improvement in activity because of its similar bond strength with hydrogen as Pt[4,7–13]. Recently, our group explored the dominant role of atomic- and Ru nanoparticles as for the HER[11], in which atomic-Ru plays a dominant role for the HER in acid electrolyte because of its appropriate H* adsorption strength, and meanwhile Ru NPs facilitate the dissociation of $H_2O$ in alkaline electrolyte. Other groups also reported a few highly efficient Ru/C HER catalysts, such as Ru nanoparticles anchored on N-doped carbon, graphene nanoplatelet, and carbon quantum dots[7,10,13]. Among the above electrocatalysts, heteroatom-doped carbon-based substrates not only had excellent electroconductivity but also showed some activities for the HER. However, the Ru nanoparticles often fall off from the carbon substrates and thus cause catalyst failure. To enhance the stability of Ru nanoparticles, transition metal oxides often were chosen as the substrates, such as $TiO_2$, $CeO_2$ and $ZrO_2$, the strong interaction between Ru nanoparticles and metal oxides can suppress detachment of catalysts from the substrates. Importantly, the interaction can tune surface electronic structure and energy level of Ru nanoparticles by the formation of Ru-O-M (M = Ti, Ce, Zr) bonds in $Ru/MO_2$ nanocomposites that were used for various thermal-catalysis reactions, such as carbon oxide methanation[14], dry reforming of methane[15] and hydrogenation of levulinic acid[16]. Huang et al. reported a Ru-doped $TiO_2$ HER electrocatalyst in an alkaline solution, in which the Ru$^{5+}$ and Ti$^{3+}$ synergistically enhanced the activity with appropriate hydrogen-adsorption Gibbs free energies[17]. In addition, the crystalline, morphology, and electronic structure of metal oxides themselves also have a profound effect on the electrocatalytic performance of $Ru/MO_2$ nanocomposites, for example, the enriched surface defects of $CeO_2$ are favorable for the formation of Ru-O-Ce bonds by Ru ions diffusing into $CeO_2$ surface lattice[18]. So far, the $HfO_2$ is seldom used as the substrate or active component in electrocatalysis because of its large bandgap. However, it had been applied in thermal catalysis. As a Lewis acid site, isolated Hf facilitates acetone conversion to isobutene[19]. Pd/$HfO_2$ has been reported to be highly active for methane combustion[20]. By constructing a composite of Ru nanoparticles supported by $HfO_2$ substrate with oxygen defect, can the surface electronic structure of Ru nanoparticles be well optimized as highly efficient HER electrocatalyst?

In this work, we demonstrate that Ru nanoparticles supported by oxygen vacancies-riched $HfO_2$ ($V_O$-Ru/$HfO_2$-OP, $V_O$, O, and P refer to oxygen vacancies, oleylamine, and polyvinyl pyrrolidone, respectively) exhibit excellent HER activity and stability in alkaline electrolytes. The Ru content is only 0.9 wt%, which greatly decreases the price of the catalyst compared with that of commercial Ru/C and Pt/C. The interaction between Ru nanoparticles and $HfO_2$ by Ru-O-Hf bonds as well as $V_O$ in the substrate synergistically promote the water dissociation. DFT calculations reveal that the $d$-band center of Ru could be tuned closer to the Fermi level owing to the synergistic effects of the Ru-O-Hf bonds and $V_O$, which is beneficial for the adsorption of water, as it lowers the energy barrier for water dissociation.

## Results and discussion

**Phase and structural characterizations.** Preparation of $V_O$-Ru/$HfO_2$-OP was conducted in two continuous steps. First, a modified polyol process with oleylamine and polyvinyl pyrrolidone as structure-directing agents was employed to prepare pristine Ru/$HfO_2$-OP. Second, $V_O$ was introduced by annealing under a $H_2$/Ar atmosphere. The primary crystalline phase in $V_O$-Ru/$HfO_2$-OP was identified as monoclinic $HfO_2$ with the lattice parameters of $a = 0.512$ nm, $b = 0.517$ nm, and $c = 0.530$ nm (PDF No. 97-005-7385) by X-ray diffraction (XRD) patterns (Fig. 1a). No diffraction peaks of Ru can be detected because of the ultralow content of Ru in the composite, which is only 0.9 wt%, as determined by inductively coupled plasma atomic emission spectrometry. Field-emission scanning electron microscopy (FESEM) shows that the $V_O$-Ru/$HfO_2$-OP nanoparticles are uniformly dispersed (Fig. 1b). Figure 1c and Supplementary Fig. 1 show the typical TEM images of $V_O$-Ru/$HfO_2$-OP, which demonstrate that the nanoparticles have a porous structure and a diameter of 60–80 nm. The high-resolution TEM (HRTEM) image shown in Fig. 1e corresponds to the region depicted in Fig. 1d, marked with a brown rectangle. The measured lattice spacing of 0.261 nm was attributed to the (002) plane of monoclinic $HfO_2$. The hexagonal close-packed (hcp) lattice with a lattice spacing of 0.234 nm is assigned to Ru nanoparticles[21]. The high-angle annular dark-field scanning transmission electron microscopy (HAADF-STEM) image shown in Fig. 1f further demonstrates the porous structure of the $V_O$-Ru/$HfO_2$-OP catalyst. The corresponding elemental mappings (Fig. 1g–i) show that Hf, O, and Ru are uniformly distributed in the nanoparticles. $V_O$-Ru/$HfO_2$-P, $V_O$-Ru/$HfO_2$-O, and pristine $HfO_2$ were also prepared following a similar synthetic procedure to that for $V_O$-Ru/$HfO_2$-OP, except for the addition of oleylamine, PVP, or RuCl$_3$·xH$_2$O. The basic physical characterizations of $V_O$-Ru/$HfO_2$-P, $V_O$-Ru/$HfO_2$-O, and pristine $HfO_2$ are shown in Supplementary Figs. 2–6. The XRD pattern of $V_O$-Ru/$HfO_2$-P shows clear diffraction peaks corresponding to the hexagonal crystal structure of Ru (PDF No. 99-000-3234) (Supplementary Fig. 3a). The average diameter of the Ru nanoparticles in $V_O$-Ru/$HfO_2$-P is 7 nm, calculated according to the Debye–Scherrer equation[22], which is comparable to the EDS elemental linear scanning result (Supplementary Fig. 5b). The larger size of Ru nanoparticles in $V_O$-Ru/$HfO_2$-P indicates the key role of oleylamine in tuning the size of the Ru nanoparticles. The addition of PVP as a stabilizer effectively prevented the aggregation of $HfO_2$ nanoparticles.

Advanced characterization techniques, including X-ray photoelectron spectroscopy (XPS), X-ray absorption near edge structure (XANES), and extended X-ray absorption fine structure (EXAFS) measurements, were employed to gain insights into the valence state and elemental composition of the prepared catalysts. The XPS survey spectrum further reveals that Hf, O, and Ru are dominant in $V_O$-Ru/$HfO_2$-OP (Supplementary Fig. 7). The XPS of $V_O$-Ru/$HfO_2$-OP depicts a Ru $3d_{3/2}$ peak, which shows a significant shift to a higher binding energy relative to that of bulk Ru (Fig. 2a). These positive core level shifts involved in the smaller metal clusters supported on less conductive substrates can be interpreted by final state effects[23,24]. As the final state of the photoemission process, the positive hole can be less efficiently screened, leading to a positive core level shift with decreasing particle size[25]. Thus, the size of the Ru cluster in the $V_O$-Ru/$HfO_2$-OP is much smaller than that of the bulk Ru. In contrast, $V_O$-Ru/$HfO_2$-P shows a negative shift of 0.4 eV compared to that of $V_O$-Ru/$HfO_2$-OP, owing to the larger Ru cluster size of $V_O$-Ru/$HfO_2$-P. The binding energy for Ru $3d_{3/2}$ of $V_O$-Ru/$HfO_2$-O is located in the middle of $V_O$-Ru/$HfO_2$-OP and $V_O$-Ru/$HfO_2$-P, demonstrating that the Ru cluster size in $V_O$-Ru/$HfO_2$-O is between those of $V_O$-Ru/$HfO_2$-OP and $V_O$-Ru/$HfO_2$-P. The smaller size of the Ru

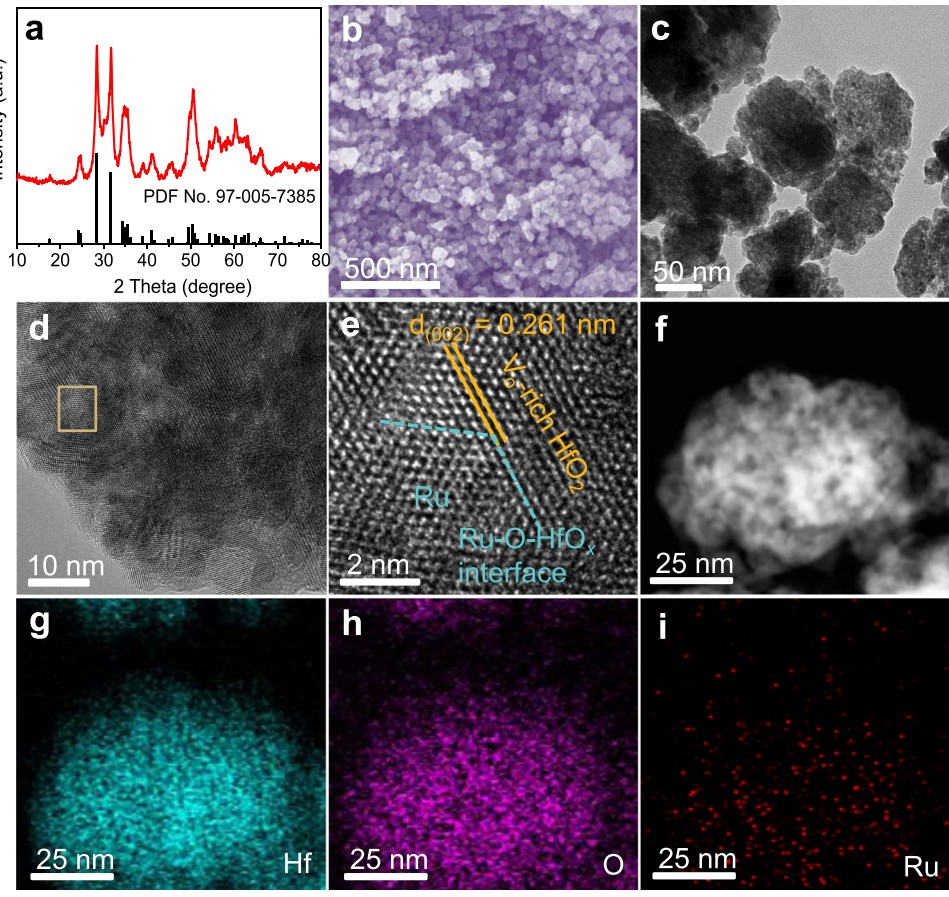

**Fig. 1 Phase and structural characterizations. a** XRD pattern, **b** SEM image, **c** TEM image, **d** HRTEM image, **e** Magnified HRTEM image, **f–i** HAADF-STEM image and the corresponding EDS elemental mappings (Hf: Olive, O: Magenta, Ru: Red).

cluster signifies more Ru-O-Hf bonds. Besides, the three peaks around at 284.6, 286.2, and 288.8 eV in spectra of C 1$s$ and Ru 3$d$ of Ru, $V_O$-Ru/HfO$_2$-O, and $V_O$-Ru/HfO$_2$-OP belong to C=C, C-O, and O-C=O, respectively, derived from the carbon contamination on the catalysts surface[26]. Meanwhile, the three peaks centered at 279.7, 280.7, and 282.4 eV in the spectra of C 1$s$ and Ru 3$d$ of $V_O$-Ru/HfO$_2$-P are attributed to Ru 3$d_{5/2}$ of Ru$^0$, Ru$^{4+}$, and Ru$^{5+}$ [17,27], respectively, indicating the possible oxidation of catalyst sample when exposed in the air. While the remaining three peaks at 284.2, 286.0, and 288.3 eV in C 1$s$ and Ru 3$d$ XPS spectra of $V_O$-Ru/HfO$_2$-P are assigned to C 1$s$ originated from adsorbed carbon species[17]. As corroborated by Fig. 2b, the high-resolution O 1$s$ of the as-synthesized $V_O$-Ru/HfO$_2$-OP catalyst presents three peaks at ~530.1, 531.3, and 532.2 eV, corresponding to lattice oxygen, oxygen vacancies, and adsorbed water molecules, respectively, demonstrating the presence of $V_O$[28,29]. A signal at $g = 2.001$ resulting from the unpaired electrons trapped by $V_O$ is detected through electron paramagnetic resonance (EPR) (Supplementary Fig. 8), which further confirms the presence of $V_O$. The binding energies of Hf 4$f_{5/2}$ and Hf 4$f_{7/2}$ core levels for $V_O$-Ru/HfO$_2$-OP are 18.4 and 16.7 eV (Supplementary Fig. 9a), respectively, which are in good agreement with the values of Hf 4$f_{5/2}$ and Hf 4$f_{7/2}$ doublet peaks for HfO$_2$[30]. No obvious shift in the O 1$s$ and Hf 4$f$ peaks of $V_O$-Ru/HfO$_2$-OP relative to that of pristine HfO$_2$ that could be attributed to the ultralow Ru loading on the HfO$_2$ support was observed.

The Ru K-edge XANES spectra of $V_O$-Ru/HfO$_2$-OP, $V_O$-Ru/HfO$_2$-O, and $V_O$-Ru/HfO$_2$-P are shown in Fig. 2c. The energy absorption threshold value of $V_O$-Ru/HfO$_2$-OP is between that of Ru foil and commercial RuO$_2$, indicating that the Ru

nanoparticles loaded on HfO$_2$-OP are positively charged. The pre-edge adsorption of the Ru K-edge for $V_O$-Ru/HfO$_2$-P negatively shifted and becomes closer to that of the Ru foil, demonstrating the relatively low oxidation state of Ru in $V_O$-Ru/HfO$_2$-P than that in $V_O$-Ru/HfO$_2$-OP. The Ru in $V_O$-Ru/HfO$_2$-O exhibits a slightly higher valence state than that in $V_O$-Ru/HfO$_2$-OP and is closer to that of RuO$_2$. The Fourier transform (FT) of the EXAFS spectra of the synthesized catalysts and references are shown in Fig. 2d. For $V_O$-Ru/HfO$_2$-OP, two scattering peaks originating from Ru-O-Hf coordination at ~1.62 Å and Ru-Ru coordination at ~2.36 Å were detected. The spectrum of $V_O$-Ru/HfO$_2$-P shows a higher intensity peak at 2.49 Å ascribed to Ru–Ru interaction and a relatively low-intensity Ru-O-Hf peak at 1.85 Å. Both peaks shifted to a higher distance compared to those of $V_O$-Ru/HfO$_2$-OP. The qualitative evaluation of the spectra implies that the intensity of the Ru-O-Hf bond in $V_O$-Ru/HfO$_2$-OP is higher than that in $V_O$-Ru/HfO$_2$-P. In contrast, the intensity of the Ru-Ru bond in the catalyst is lower than that of $V_O$-Ru/HfO$_2$-P[31]. In other words, the number of Ru-O-Hf bonds in $V_O$-Ru/HfO$_2$-OP is greater than that in $V_O$-Ru/HfO$_2$-P. For $V_O$-Ru/HfO$_2$-O, the locations of the Ru-O-Hf and Ru-Ru bonds are similar to those of $V_O$-Ru/HfO$_2$-OP, except for a slight shift of Ru-O-Hf to a lower distance (1.60 Å) and Ru-Ru to a higher distance (2.44 Å). The XANES spectra for the Hf L$_3$-edge of pristine HfO$_2$, $V_O$-Ru/HfO$_2$-OP, $V_O$-Ru/HfO$_2$-O, and $V_O$-Ru/HfO$_2$-P are shown in Supplementary Fig. 9b. The white line peak position of Hf L$_3$-edge XANES for $V_O$-Ru/HfO$_2$-OP is located at the same position as that of pristine HfO$_2$. Moreover, the higher white line suggests that Hf in the $V_O$-Ru/HfO$_2$-OP composite possesses more empty $d$-orbital states and thus less electron

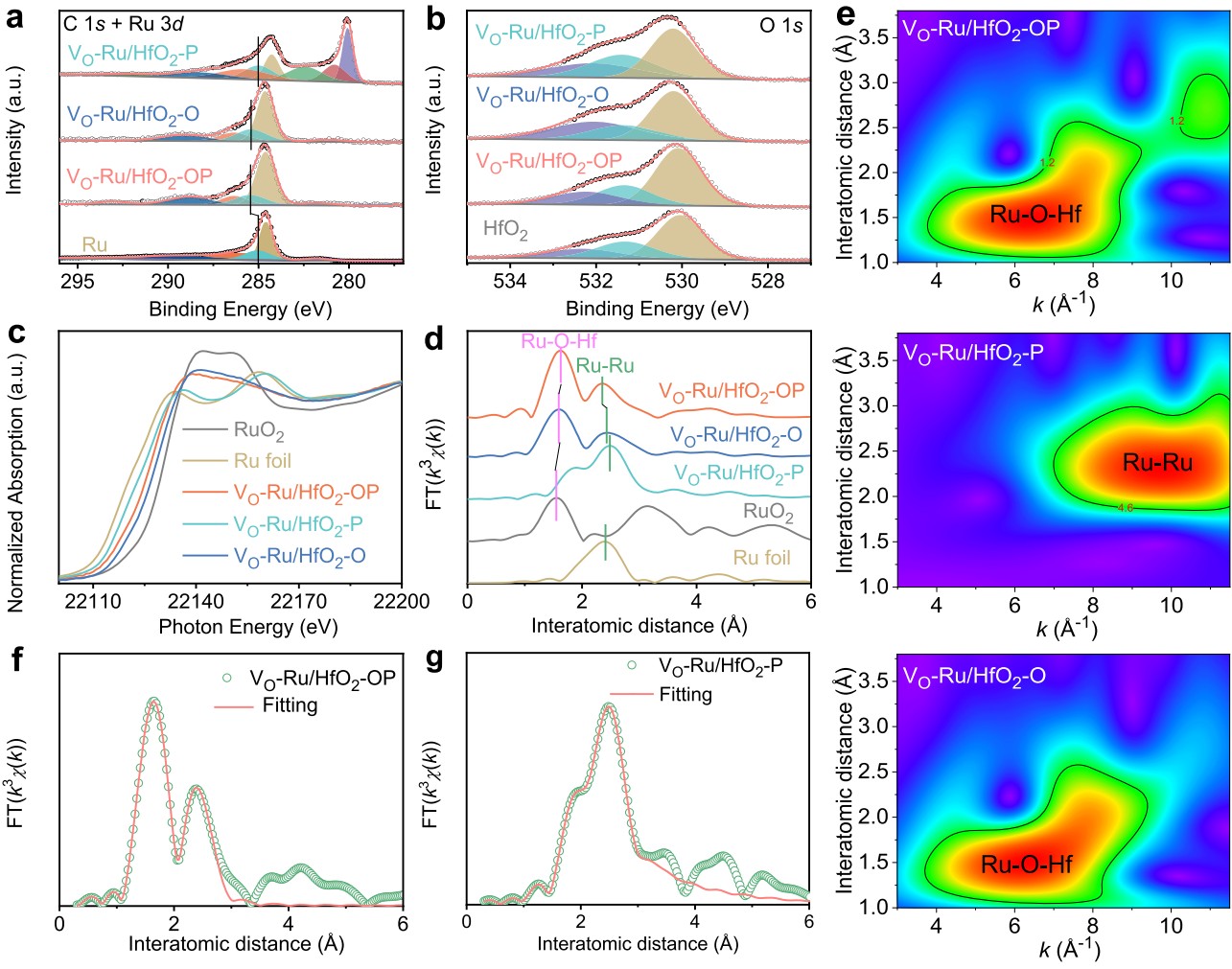

**Fig. 2 Electronic and fine structural characterizations. a** High-resolution XPS spectra of Ru 3$d$ for Ru powder, V$_O$-Ru/HfO$_2$-OP, V$_O$-Ru/HfO$_2$-O, and V$_O$-Ru/HfO$_2$-P. **b** High-resolution XPS spectra of O 1$s$ for HfO$_2$, V$_O$-Ru/HfO$_2$-OP, V$_O$-Ru/HfO$_2$-O, and V$_O$-Ru/HfO$_2$-P. **c** Ru K-edge XANES spectra and **d** Fourier transforms of the Ru K-edge EXAFS spectra of Ru foil, RuO$_2$, V$_O$-Ru/HfO$_2$-OP, V$_O$-Ru/HfO$_2$-O, and V$_O$-Ru/HfO$_2$-P. **e** WT of V$_O$-Ru/HfO$_2$-OP, V$_O$-Ru/HfO$_2$-P, and V$_O$-Ru/HfO$_2$-O, respectively. FT-EXAFS fitting curves of **f** V$_O$-Ru/HfO$_2$-OP and **g** V$_O$-Ru/HfO$_2$-P.

density[32]. The corresponding FT curves of the above four catalysts are shown in Supplementary Fig. 9c.

To further explore changes in the electronic structure and valence state, wavelet transform (WT) with high resolution in both k and R space analyses were carried out. Figure 2e shows the WT EXAFS contour plots of V$_O$- Ru/HfO$_2$-OP, V$_O$-Ru/HfO$_2$-O, and V$_O$-Ru/HfO$_2$-P. The WT EXAFS contour plots of commercial Ru and RuO$_2$ are shown in Supplementary Fig. 10. The maximum-intensity value at $k \approx 6.5\,\text{Å}^{-1}$ ascribed to Ru-O-Hf backscattering contributions is clearly detected for V$_O$-Ru/HfO$_2$-OP and V$_O$-Ru/HfO$_2$-O. In contrast, the Ru-Ru WT signal of V$_O$-Ru/HfO$_2$-OP and V$_O$-Ru/HfO$_2$-O is very weak, which further reveals the increase in Ru-O-Hf bonds and decrease in Ru-Ru bonds tuned by oleylamine surfactants. In contrast, no obvious WT signal could be detected for Ru-O-Hf bonds of V$_O$-Ru/HfO$_2$-P in the lower coordination shell; however, a strong WT signal near $9.6\,\text{Å}^{-1}$ corresponding to Ru-Ru contribution was observed. Quantitative EXAFS curve fitting for both R and k space was carried out to determine the structural parameters, as shown in Supplementary Table 1; the corresponding fitting results are shown in Fig. 2f, g and Supplementary Fig. 11. The local structural parameters further demonstrate the stronger Ru-O-Hf bonds and weaker Ru-Ru bonds of V$_O$-Ru/HfO$_2$-OP compared to those in V$_O$-Ru/HfO$_2$-P, which might favor a superior HER catalytic activity.

**Activity and stability evaluation**. The catalytic properties of the as-synthesized V$_O$-Ru/HfO$_2$ series were investigated in a typical three-electrode setup using 1.0 M KOH solution as the electrolyte. Commercial Pt/C and Ru/C were used as the references. The typical polarization curves of HfO$_2$, V$_O$-Ru/HfO$_2$-OP, V$_O$-Ru/HfO$_2$-O, V$_O$-Ru/HfO$_2$-P, commercial Ru/C (Ru: 5 wt%), and Pt/C (Pt: 20 wt%) at a scan rate of 5 mV s$^{-1}$ are presented in Fig. 3a. Impressively, V$_O$-Ru/HfO$_2$-OP demonstrated substantially better catalytic activity than Ru/C, V$_O$-Ru/HfO$_2$-O, and V$_O$-Ru/HfO$_2$-P, indicating that a higher number of Ru-O-Hf bonds is critical to increase the HER catalytic performance. Nevertheless, pristine HfO$_2$ is HER-inert with a negligible current, even at a high applied potential. The measured overpotential corresponding to 10 mA cm$^{-2}$ is 39, 79, 90, and 145 mV for V$_O$-Ru/HfO$_2$-OP, Ru/C, V$_O$-Ru/HfO$_2$-O, and V$_O$-Ru/HfO$_2$-P (Supplementary Fig. 12), respectively. As a result, V$_O$-Ru/HfO$_2$-OP exhibited the best catalytic activity among the investigated samples and was even close to that of state-of-the-art Pt/C. Figure 3b illustrates the Tafel slopes based on the corresponding LSV curves shown in Fig. 3a. The values are 22, 29, 44, 66, and 133 mV dec$^{-1}$ for Pt/C, V$_O$-Ru/HfO$_2$-OP, Ru/C, V$_O$-Ru/HfO$_2$-O, and V$_O$-Ru/HfO$_2$-P, respectively. The lower Tafel slope of V$_O$-Ru/HfO$_2$-OP with a higher number of Ru-O-Hf bonds highlights the effective facilitation of the hydrogen evolution kinetics. The V$_O$-Ru/HfO$_2$-OP also showed ultra-high mass activity (A g$_{noble\ metal}^{-1}$ normalized by

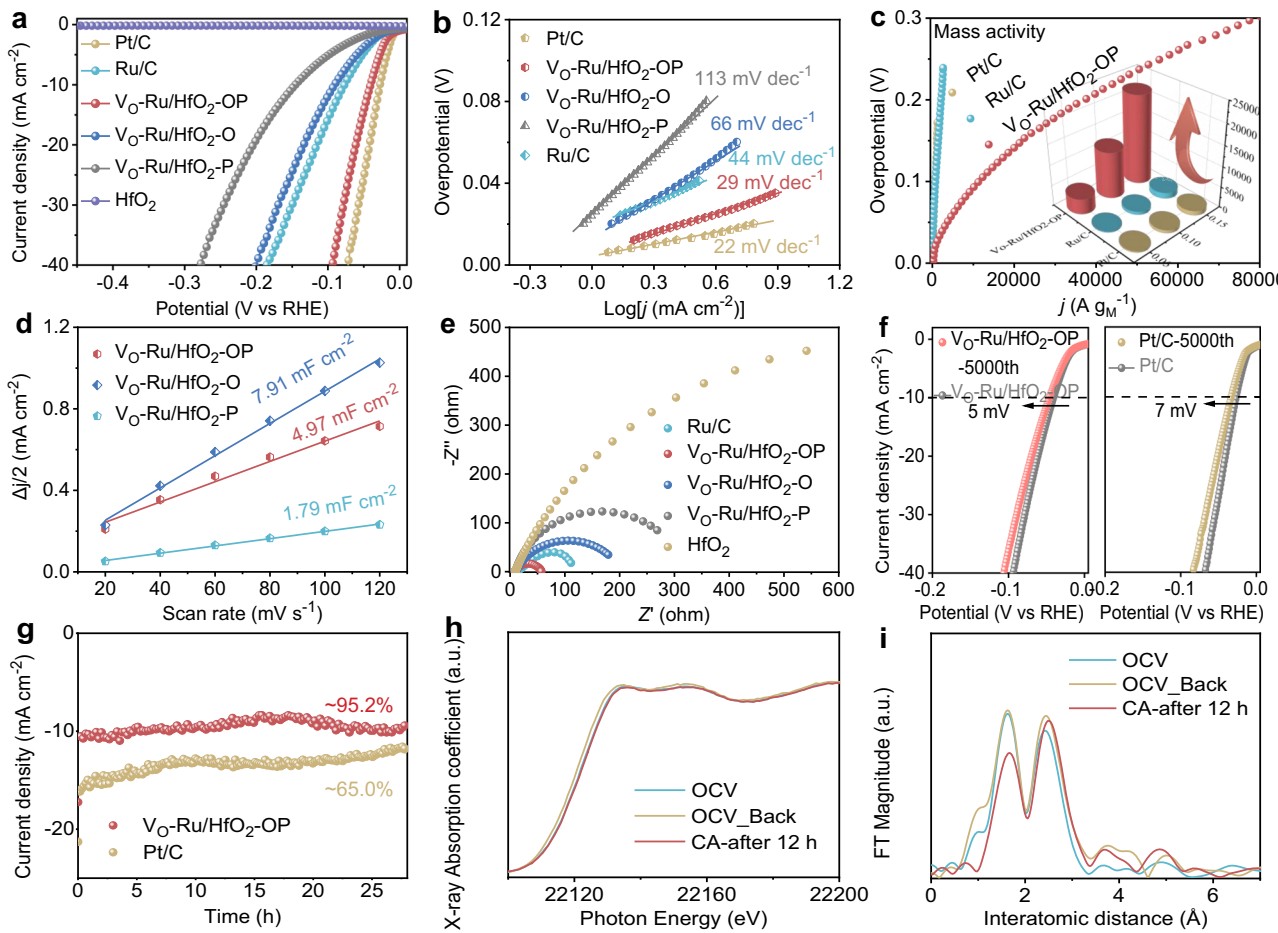

**Fig. 3 HER performance in 1.0 M KOH. a** The polarization curves of $HfO_2$, $V_O$-Ru/$HfO_2$-OP, $V_O$-Ru/$HfO_2$-O, $V_O$-Ru/$HfO_2$-P, and commercial Ru/C (Ru: 5 wt%), Pt/C (Pt: 20 wt%). **b** Tafel slopes of $V_O$-Ru/$HfO_2$-OP, $V_O$-Ru/$HfO_2$-O, $V_O$-Ru/$HfO_2$-P, Ru/C, and Pt/C. **c** Mass activities normalized by the noble metal mass. **d** Capacitive $\Delta j/2$ as a function of the scan rate for $V_O$-Ru/$HfO_2$-OP, $V_O$-Ru/$HfO_2$-O, $V_O$-Ru/$HfO_2$-P. **e** Nyquist plots of $HfO_2$, $V_O$-Ru/$HfO_2$-OP, $V_O$-Ru/$HfO_2$-O, $V_O$-Ru/$HfO_2$-P, and Ru/C. **f** The polarization curves of Pt/C and $V_O$-Ru/$HfO_2$-OP before and after 5000 CV cycles. **g** The stability tests for Pt/C and $V_O$-Ru/$HfO_2$-OP at a constant potential of −0.039 V (vs. RHE) for 28 h. The XC-72 was used as conductive support in all measurement of $HfO_2$. **h** *Operando* Ru K-edge XANES spectra and **i** corresponding Fourier-transformed (FT) magnitudes in *operando* Ru K-edge EXAFS spectra of $V_O$-Ru/$HfO_2$-OP before and after CA testing at −0.039 V (vs. RHE) for 12 h.

the mass of noble metal), which is ~20 times and 17 times higher than those of commercial Pt/C and Ru/C, respectively, at an over-potential of 0.1 V (Fig. 3c). A series of CVs were employed to study the effect of electrochemically active surface areas on the intrinsic activities of the Ru/$HfO_2$ series (Supplementary Fig. 13). As illustrated in Fig. 3d, the electrochemical double-layer capacitance ($C_{dl}$) of $V_O$-Ru/$HfO_2$-OP increased to 2.8 times higher than that of $V_O$-Ru/$HfO_2$-P, although it is still smaller than that of $V_O$-Ru/$HfO_2$-O. However, the ECSA-normalized specific current density of $V_O$-Ru/$HfO_2$-OP is 8 times and 2.8 times higher than those of $V_O$-Ru/$HfO_2$-O and $V_O$-Ru/$HfO_2$-P (Supplementary Fig. 14) at a potential of -0.039 V (vs. RHE), respectively, demonstrating the considerably higher number of active sites as well as improved intrinsic catalytic activity, synergistically resulting in enhanced HER performance.

The electrochemical impedance spectroscopy (EIS) curves shown in Fig. 3e display a smaller charge transfer resistance ($R_{ct}$) of $V_O$-Ru/$HfO_2$-OP (49.1 ohm) than that of Ru/C (116.5 ohm), $V_O$-Ru/$HfO_2$-O (200.1 ohm), $V_O$-Ru/$HfO_2$-P (301.7 ohm) (Supplementary Fig. 15 and Supplementary Table 2), and pristine $HfO_2$ (4276.0 ohm), suggesting the facilitated electron transfer and thus faster electrocatalytic kinetics for HER[33,34]. The increased over-potential value is merely 5 mV at a current density of 10 mV cm$^{-2}$ for $V_O$-Ru/$HfO_2$-OP after continuous 5000 CV cycles, which is superior to that of Pt/C (7 mV) (Fig. 3f). The chronoamperometry

(CA) test results further confirmed the better long-term durability of $V_O$-Ru/$HfO_2$-OP than that of Pt/C. No obvious current attenuation can be observed for $V_O$-Ru/$HfO_2$-OP after continuous testing at a benchmark of 10 mA cm$^{-2}$ for 28 h (Fig. 3g). The Fig. 3h and i show the *operando* Ru K-edge XANES spectra and corresponding Fourier-transformed (FT) magnitudes in *operando* Ru K-edge EXAFS spectra of $V_O$-Ru/$HfO_2$-OP before and after CA testing at −0.039 V (vs. RHE) for 12 h. Evidently, both the XANES and EXAFS are similar to the initial open-circuit voltage (OCV) ones when the applied potential returned to OCV after long-term CA testing, indicating the high stability. The optimal synthetic conditions, including the optimal molar ratio of the raw material of Ru to Hf, the optimal calcination temperature, the PVP dosage, and the ratio of O (oleylamine) to P (PVP), were systematically studied. Evidently, the $V_O$-Ru/$HfO_2$-OP catalyst prepared with a molar ratio of Ru to Hf of 1:1, an annealing temperature of 750 °C, the 50 mg PVP, and the ratio of O to P of 4: 50 showed the best electrocatalytic activity for HER (Supplementary Figs. 16–19).

**In situ and *operando* XAS analysis of $V_O$-Ru/$HfO_2$-OP.** In order to monitor the electronic state of the Ru active sites during the HER, potential-dependent Ru K-edge XAS measurements were performed using a home-made *operando* three-electrode cell

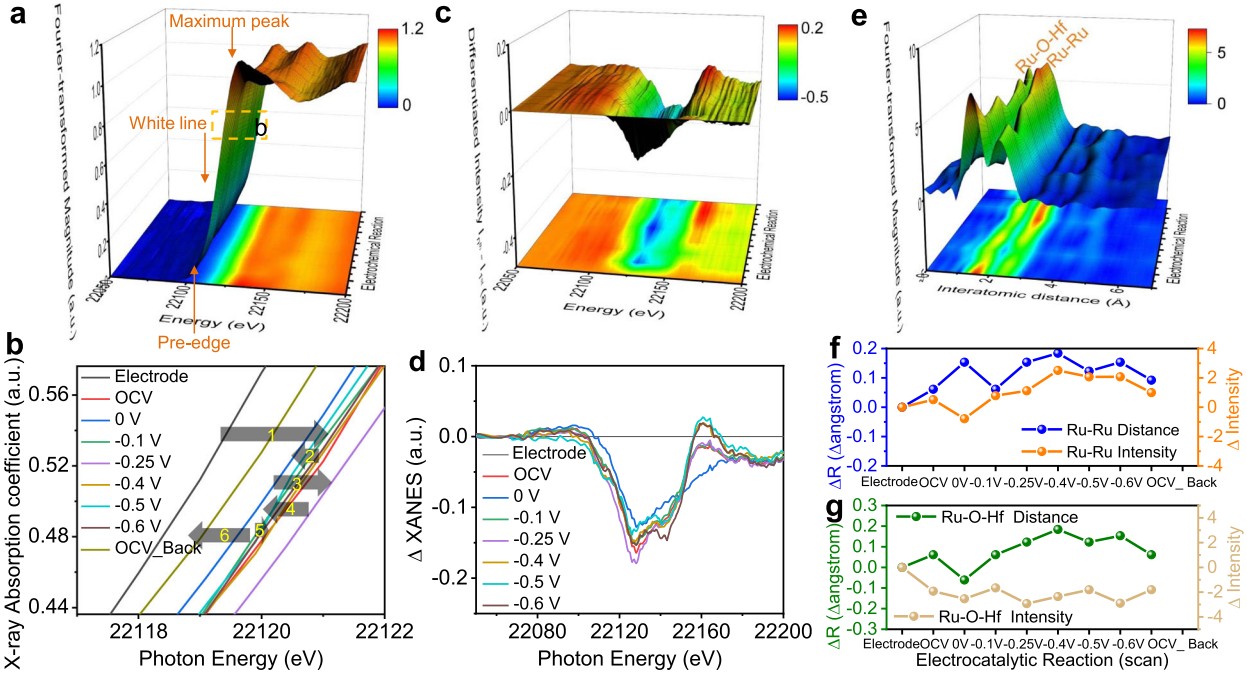

**Fig. 4** *Operando* **Ru K-edge XANES and EXAFS spectra of V$_O$-Ru/HfO$_2$-OP. a** Three-dimensional plot of *operando* Ru K-edge XANES spectrum recorded at varied potential from OCV to −0.6 V (vs. RHE) during the HER catalysis. **b** The reversible change of Ru valence state during the electrocatalytic HER process: 1 refers to electrode – OCV: oxidation; 2 refers to OCV – 0 V: reduction; 3 refers to 0 V – −0.1 V – −0.25 V: oxidation; 4 refers to −0.25 V – −0.4 V – −0.5 V: reduction; 5 refers to −0.5 V – −0.6 V: oxidation; 6 refers to −0.6 V – OCV back: reduction. **c** Three-dimensional plot and **d** Curves of normalized differentiated XANES intensity (I$_{nth}$ – I$_{1st}$) in *operando* Ru K-edge XANES spectra. **e** Three-dimensional plot of *operando* Ru K-edge FT-EXAFS spectrum of V$_O$-Ru/HfO$_2$-OP. Changes in the distances and intensities of **f** Ru-Ru and **g** Ru-O-Hf in the *operando* Ru K-edge FT-EXAFS spectrum of V$_O$-Ru/HfO$_2$-OP.

system. Figure 4a, b and Supplementary Fig. 20 present the *operando* Ru K-edge XANES spectra recorded at different potential from open-circuit condition to −0.6 V (vs. RHE). Three important peaks labeled as pre-edge peak (1 s → 4d transition), white line peak (1 s → 5p transition), and maximum peak (1 s → 5p transition multiple scattering), are obviously changed along with the applied potentials. The variation can be more clearly discerned from the differentiated Ru K-edge XANES intensity (I$_{nth}$ – I$_{1st}$) in Fig. 4c, d. The change of white line energy signifies the oxidation number variation, while the change of pre-edge peak and maximum peak is relevant to the degree of structural distortion. From the ex situ sample to the OCV, a positive shift of absorption edge towards higher energy was occurred, accompanied by an intensity increase of white line peak, implying the increased oxidation state of Ru[35]. While, when cathodic potential of 0 V was applied, the absorption edge of Ru K-edge XANES spectrum was shifted to lower energy compared with the case under the OCV, along with the decreased intensity of white line, meaning a decrease of Ru oxidation state. Further to switch voltage to −0.1 V (vs. RHE) and −0.25 V (vs. RHE), the adsorption edge of Ru XANES spectra was shifted back to higher energy in relation to that under 0 V condition, and the white line intensity was also increased, demonstrating the increase of Ru oxidation state. If the more negative voltages of −0.4 and −0.5 V were applied, the oxidation state of Ru went down again evidenced by the negative shift of adsorption edge and decreased white line intensity. Such reversible redox occurs in this way until the applied voltage back to the OCV condition. The reduction of Ru oxidation state demonstrates the electrons transfer from intermediates to Ru, which is benefit for attachment of H intermediates; While the increase of Ru oxidation state means the electrons transfer from Ru to intermediates, which is favors of the detachment of H intermediates[36]. Thus, vigorous oxidation and reduction reactions induce the vigorous oxidation number change

of Ru, making that the intermediates are easily attached and detached. As evidenced by the variations of pre-edge peaks and maximum peaks, the structural change occurred from the ex situ sample to the OCV stage, which corresponds to the electrode activation process. Moreover, the structural change was occurred continuously during the HER, but it is reversibly and stable. The above results indicate that the V$_O$-Ru/HfO$_2$-OP is flexible with respect to structural distortions and the reversible redox reaction of Ru, resulting in high catalytic activity as well as stability.

Figure 4e and Supplementary Fig. 21 display the *operando* Ru K-edge FT-EXAFS spectra. The two main FT peaks are directly related to interatomic distances, attributing to Ru-O-Hf and Ru-Ru bonds, respectively. Apparently, the FT peak positions and intensity of Ru-O-Hf and Ru-Ru underwent a marked change during the HER catalysis. The bonds of Ru-O-Hf and Ru-Ru were contracted and stretched during the reaction, and the FT peaks intensity were increased and decreased, as more clearly presented in Fig. 4f, g. Impressively, the change frequency of interatomic distances relative to Ru-O-Hf and Ru-Ru is high, however, the variation of interatomic distance changes is low, effectively demonstrating the flexible structure of V$_O$-Ru/HfO$_2$-OP and highly stable it during the alkaline hydrogen electrocatalysis, which is consistent with the results of *operando* XANES spectra. Besides, the change frequency of FT peak intensity is high, but it is low for variation of intensity. This is due to the fast adsorption and desorption rate of the intermediate[36,37]. Thus, intermediate species are easily absorbed and desorbed at Ru-O-Hf and Ru-Ru sites, bringing about fast reaction kinetics.

**Density functional theory calculations.** Spin-polarized DFT calculations implemented in the Vienna ab initio simulation package (VASP) were performed to gain a better understanding of the enhanced performance of V$_O$-Ru/HfO$_2$-OP for HER in alkaline electrolytes. The experimental results showed that the

HER performance can be improved by increasing the number of Ru-O-Hf bonds in the Ru/HfO$_2$ series. The size of the Ru nanoparticles is inversely proportional to the number of Ru-O-Hf bonds; thus, the model system for the active sites could use Ru nanoparticles with different sizes. Consequently, correlative theoretical models including Ru (001), HfO$_2$ (001), Ru$_3$, Ru$_6$, Ru$_{10}$, and Ru$_{13}$ clusters, and supported Ru clusters denoted as Ru$_3$/HfO$_2$, V$_O$-Ru$_3$/HfO$_2$, Ru$_6$/HfO$_2$, Ru$_{10}$/HfO$_2$, and Ru$_{13}$/HfO$_2$ were constructed, as shown in Supplementary Figs. 22, 23. Previous ab initio thermodynamic phase diagrams show that the (001) face is indeed a thermodynamically stable face of HfO$_2$[38]. The O-terminated (001) surface is the most stable surface for HfO$_2$, as revealed by total energy-based DFT calculations (Supplementary Fig. 24). Thus, the O-terminated (001) plane of HfO$_2$ was selected as the substrate for loading Ru clusters with different numbers of Ru-O-Hf bonds. In addition, to better understand the effect of oxygen vacancies, HfO$_2$ without and with O defects, the position of V$_O$ localizing, as well as V$_O$ concentration were also taken into account. As revealed by total energy-based DFT calculation, the oxygen vacancy localized on the surface of HfO$_2$ is the most stable (Supplementary Fig. 25). The calculated most stable adsorption structures of Ru$_3$ on HfO$_2$ and V$_O$-HfO$_2$ (HfO$_2$ with one O defect and the V$_O$ concentration is 1.56%) are shown in Fig. 5a, b and Supplementary Fig. 26a, respectively. The adsorption energy of Ru$_3$ on V$_O$-HfO$_2$ ($-7.20$ eV) is higher than that on HfO$_2$ ($-5.63$ eV), and hence, the Ru clusters supported on oxygen-deficient HfO$_2$ is more stable. To further increase V$_O$ concentration, a model of HfO$_2$ with double O defect (V$_{2O}$-HfO$_2$), and the concentration of V$_O$ is 3.12% were constructed (Supplementary Fig. 27). The larger adsorption energy of Ru$_3$ on V$_{2O}$-HfO$_2$ ($-8.60$ eV) than that on V$_O$-HfO$_2$ ($-7.20$ eV), demonstrating that the Ru clusters supported on V$_{2O}$-HfO$_2$ is more stable. The interaction between the metal and the support plays a very important role in controlling the catalysis of supported metal catalysts[39]. A net stronger electron transfer of 0.28 e from the Ru$_3$ cluster to defective HfO$_2$ was revealed by charge density difference analysis, which is an effective method for visualizing the charge transfer between different components as well as the bonding structures of a catalyst (Fig. 5c). Moreover, the projected density of states calculation indicates a strong orbital overlap between Ru 4$d$, Hf 5$d$, and O 2$p$ orbitals for V$_O$-Ru/HfO$_2$-OP (Fig. 5d), effectively demonstrating the strong interaction between Ru and the Vo-HfO$_2$ substrate.

In alkaline media, the overall HER reaction pathways include the dissociation of H$_2$O and the formation of adsorbed hydrogen intermediates (H$_2$O + e$^-$ + * → H* + OH$^-$), as well as the final hydrogen generation (H* + e$^-$ → 1/2H$_2$)[40–42]. Therefore, superior alkaline HER electrocatalysts should simultaneously possess moderate H binding energy and a relatively low H$_2$O dissociation barrier. Figure 5e and Supplementary Fig. 28 show the calculated adsorbed free energy ($\Delta G_{H*}$) of the hydrogen intermediate on the active sites of Ru (001) ($-0.32$ eV), Ru$_3$/HfO$_2$ ($-0.37$ eV), V$_O$-Ru$_3$/HfO$_2$-OP ($-0.34$ eV), V$_{2O}$-Ru$_3$/HfO$_2$-OP ($-0.31$ eV), Ru$_6$/HfO$_2$ ($-0.31$ eV), Ru$_{10}$/HfO$_2$ ($-0.29$ eV), and Ru$_{13}$/HfO$_2$ ($-0.35$ eV), which varied from $-0.37$ to $-0.29$ eV, indicating favorable energetics for hydrogen adsorption and desorption to form H$_2$ from all Ru-based catalysts[43]. However, the $\Delta G_{H*}$ value was 1.12 eV for HfO$_2$(001) (Supplementary Fig. 29a). Such high free energy for hydrogen adsorption hinders the formation of the H intermediate[11,44], resulting in HER-inert pristine HfO$_2$. Except for $\Delta G_{H*}$, the faster kinetics of water dissociation is a prerequisite for hydrogen evolution in alkaline electrolytes, which directly determines the HER activity[45]. The computed adsorption energy of H$_2$O for Ru(001), Ru$_3$/HfO$_2$, Ru$_6$/HfO$_2$, Ru$_{10}$/HfO$_2$, and Ru$_{13}$/HfO$_2$ were $-0.48$, $-0.78$, $-0.71$, $-0.68$, and $-0.69$ eV, respectively (Fig. 5f and Supplementary Fig. 30), demonstrating the substantially stronger

binding of water molecules to Ru/HfO$_2$ catalysts than those of Ru(001), and unsupported Ru clusters. Moreover, the calculated adsorption energy of H$_2$O is $-0.21$ eV for HfO$_2$(001) and $-0.25$ eV for V$_O$-HfO$_2$ (Supplementary Fig. 31), indicating the ignorable effect of V$_O$ on the adsorption of water. However, it is worth noting that the water adsorption energy of V$_O$-Ru$_3$/HfO$_2$ was $-0.89$ eV, it was even lower to $-1.16$ eV for V$_{2O}$-Ru/HfO$_2$ (Supplementary Fig. 27c); in fact, it is the lowest among all the Ru/HfO$_2$ series. These results suggest that the V$_O$ do not directly participate in the adsorption of water but play a primary role in perturbing the electron distribution of the Ru cluster. As reported in a previous study[46], the adsorption energy and dissociative kinetic barrier of H$_2$O have a linear Brønsted–Evans–Polanyi (BEP) relationship. Thus, the adsorption energy of H$_2$O can be used as an activity descriptor for the kinetic barrier of water dissociation. As shown in Supplementary Fig. 32, the energy barrier of water dissociation for Ru (001) is 0.77 eV, which is higher than that of HfO$_2$-supported Ru$_x$ ($x = 3$, 6, 10, 13) (Fig. 5g and Supplementary Figs. 33–36). In Ru$_3$/HfO$_2$, the energy barrier for water dissociation is 0.65 eV; hence, the Ru site in Ru/HfO$_2$ is more effective in cleaving HO-H bonds than that in Ru (001). Notably, the energy barrier is even reduced to 0.62 eV for V$_O$-Ru$_3$/HfO$_2$-OP (Supplementary Fig. 37), and 0.54 eV for V$_{2O}$-Ru/HfO$_2$-OP (Supplementary Fig. 38), suggesting that the HER activity of Ru/HfO$_2$ can be enhanced by introducing V$_O$ and the V$_O$ concentration also significantly influences the HER activity (Supplementary Table 3). By considering all steps of H$_2$ evolution under alkaline conditions, we can conclude that Ru/HfO$_2$-OP with O defect has the optimized energies for the dissociation of water and adsorption of hydrogen, as well as for the desorption of hydrogen to form H$_2$.

The differential charge density analysis shown in Fig. 5h shows that more charge transfer occurs from the Ru sites of V$_O$-Ru$_3$/HfO$_2$ (0.23 | e |) to the O atom of adsorbed H$_2$O than that of Ru (001) (0.15 | e |). Such charge transfers elongate the H-O bond from 0.96 Å in free H$_2$O to 0.98 Å in adsorbed H$_2$O, making the H$_2$O molecule activated and easier to split. Figure 5i shows the PDOS of adsorbed H$_2$O and the 4$d$ orbital of the Ru atom with the corresponding 4$d$-band center. Evidently, the $d$-band center of V$_O$-Ru$_3$/HfO$_2$ is at $-0.98$ eV, which is closer to the Fermi level compared to those of Ru$_3$/HfO$_2$ ($-1.04$ eV) and Ru (001) ($-1.50$ eV). The upward shift of the Ru $d$-band center of V$_O$-Ru$_3$/HfO$_2$ can decrease the occupation of antibonding states and lead to strong binding to H$_2$O[47], resulting in an increased adsorption energy of H$_2$O. The integrated-crystal orbital Hamilton population (ICOHP) value of Ru and adsorbed O atom in H$_2$O is $-1.90$ eV for V$_O$-Ru$_3$/HfO$_2$ (Fig. 5j), which is lower than that of Ru-O in Ru$_3$/HfO$_2$ ($-1.86$) and Ru (001) ($-1.46$), further demonstrating the stronger bonding between the active-surface Ru and adsorbed H$_2$O in V$_O$-Ru$_3$/HfO$_2$. These results indicate that water can be captured at a faster rate to facilitate the Volmer reaction on the V$_O$-Ru$_3$/HfO$_2$ surfaces.

Overall, the Ru supported on the HfO$_2$ catalyst with more Ru-O-Hf bonds and V$_O$ could significantly reduce the energy barrier for breaking the H-OH bond to accelerate water dissociation. In addition, strong metal–support interactions result in optimized energy for hydrogen adsorption and desorption. These phenomena synergistically rationalize the enhanced activity and favorable kinetics of V$_O$-Ru/HfO$_2$-OP for catalytic hydrogen evolution in alkaline electrolytes.

In summary, we developed a highly efficient electrocatalyst composed of Ru nanoparticles with Vo-HfO$_2$ for the HER in an alkaline electrolyte. The interaction between Ru nanoparticles and HfO$_2$ is a key factor in determining the HER activity. A series of Ru/HfO$_2$ catalysts were purposely prepared by choosing different surfactants to tune the number of Ru-O-Hf bonds. DFT calculations and experimental results demonstrate that the HER activity of

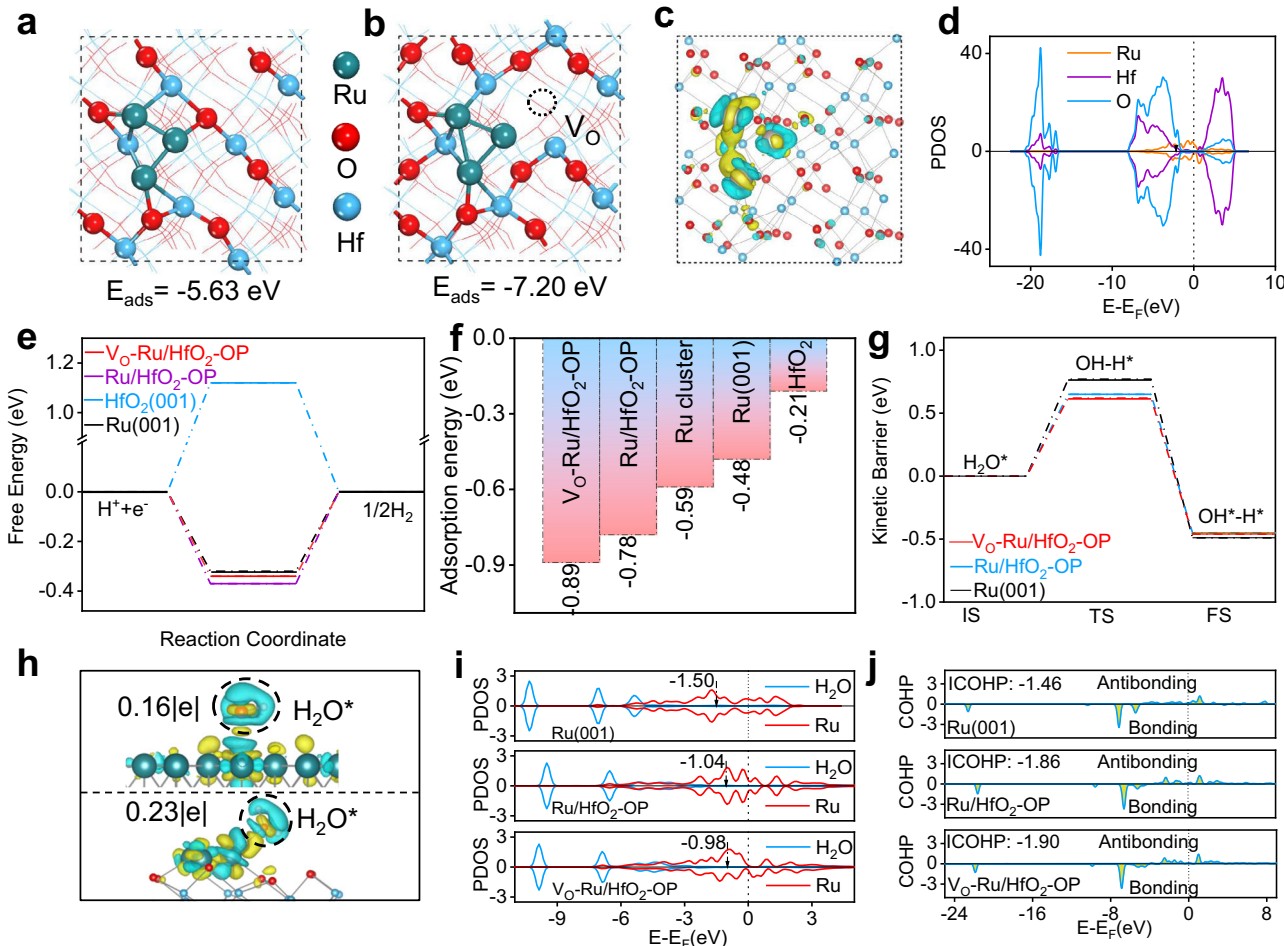

**Fig. 5 DFT calculations.** The most stable structure and adsorption energy of the Ru$_3$ cluster adsorbed on **a** HfO$_2$-OP and **b** V$_O$-HfO$_2$-OP. **c** The differential charge density distributions between Ru$_3$ clusters and V$_O$-HfO$_2$ with the isovalue of 0.001 e Å$^{-3}$. Yellow represents positive charges and olive represents negative charges. **d** The Projected density of states (PDOS) of Ru, Hf, and O atoms at V$_O$-Ru/HfO$_2$-OP. **e** The Gibbs free energy diagrams for hydrogen evolution reaction (HER) relative to standard hydrogen electrode, **f** water adsorption energy and **g** kinetic barrier of water dissociation on the active sites of different catalysts. IS, TS, and FS represent initial, transition state, and final state, respectively. **h** Differential charge density distributions between adsorbed H$_2$O and catalysts for Ru (001) (up) and V$_O$-Ru/HfO$_2$-OP (down) with the isovalue of 0.002 e Å$^{-3}$. Yellow represents positive charges and olive represents negative charges. **i** The PDOS of adsorbed H$_2$O and the 4$d$ orbital of Ru atom that directly involved in HER for Ru (001), Ru/HfO$_2$-OP, and V$_O$-Ru/HfO$_2$-OP, with corresponding Ru 4$d$-band center denoted by dash lines. **j** Crystal Orbital Hamilton population (COHP) of active Ru atom and adsorbed O atom for Ru (001), Ru/HfO$_2$-OP, and V$_O$-Ru/HfO$_2$-OP.

Vo-Ru/HfO$_2$-OP can be enhanced by controlling the number of Ru-O-Hf bonds. At the same time, V$_O$ also plays a key role in promoting HER activity. The strong metal–support interactions via Ru-O-Hf bonds and introduced V$_O$ could significantly reduce the energy barrier for breaking the H-OH bond to accelerate water dissociation. The study results can be used to improve the design and fabrication of high-performance catalysts for application in various renewable energy-conversion devices.

## Methods

**Synthesis of V$_O$-Ru/HfO$_2$-OP catalysts.** About 0.25 mmol of RuCl$_3$·xH$_2$O, 0.25 mmol of HfCl$_4$ were mixed in 60 mL of ethylene glycol under vigorous stirring. Then, 4 mL of oleylamine and 50 mg of PVP were added to the above solution. After stirring for 1 h, the reactor was flushed by Ar gas for 30 min to absolutely exhaust the air. Afterward, the solution was heated rapidly to 200 °C and maintained for 3 h under Ar flowing. When the reaction was completed, the resultant products were collected and fully washed two times with ethanol and two times with cyclohexane. Thereafter, the products were vacuum dried at 60 °C for 4 h and then, annealed at 750 °C for 2 h under H$_2$/Ar (H$_2$: 5%) atmosphere with a heating rate of 5 °C min$^{-1}$. The prepared catalyst was labeled as V$_O$-Ru/HfO$_2$-OP and directly used for electrochemical measurements.

**Synthesis of V$_O$-Ru/HfO$_2$-O and V$_O$-Ru/HfO$_2$-P catalysts.** The catalysts of V$_O$-Ru/HfO$_2$-O and V$_O$-Ru/HfO$_2$-P were prepared using a similar procedure with that of V$_O$-Ru/HfO$_2$-OP, but without adding PVP or oleylamine, respectively.

**Synthesis of HfO$_2$ catalyst.** The catalyst of HfO$_2$ was prepared using a similar procedure with that of V$_O$-Ru/HfO$_2$-OP, but without adding the RuCl$_3$·xH$_2$O.

## Data availability

The data that support the findings of this study are available from https://figshare.com/s/2023b344bfc31c2ecab6. Source data are provided with this paper.

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

## Acknowledgements

This work was supported by the Taishan Scholar Program of Shandong Province, China (ts201712045); National Natural Science Foundation of China (22102079); Shandong Provincial Key Research and Development Program (SPKR&DP) (2019GGX102069); Natural Science Foundation of Shandong Province of China (ZR2021YQ10 and ZR2018BB008); Doctoral Found of QUST (010029081 and 010029075) and 2019 Research Funds (1.190002.01) of Ulsan National Institute of Science and Technology (UNIST).

## Author contributions

X.L. and J.C. proposed the research and designed the experiments. Q.Q. and X.L. supervised the project, wrote the manuscript, and performed the analysis. S.L. carried out the DFT calculations. G.L. and Z.L. carried out the experiments. H.J. and M.G.K. conducted XAS and other physical characterizations.

## Competing interests

The authors declare no competing interests.
