## [Peer Review File · Nature Communications]

The Synergistic Effect of Hf-O-Ru Bonds and Oxygen Vacancies in Ru/HfO₂ for Enhanced Hydrogen EvolutionREVIEWER COMMENTS

Reviewer #1 (Remarks to the Author):

In this manuscript the authors have synthesized different types of Ru/HfO₂ electrocatalysts for the hydrogen evolution reaction (HER) in alkaline electrolyte. The results suggest that the HER activity can be affected by the formation of the Ru-O-Hf bonds and by the presence of oxygen vacancies. However, there are some problems in the manuscript. The authors should address the following issues before the manuscript can be considered for publications in any journal.

1. Similar to many recent electrocatalysis papers, the authors attempt to combine the experimental HER results with XANES/EXAFS characterization and DFT calculations. This combined approach can be quite useful if it is done with expertise. However, in the current manuscript the three components are poorly integrated. Below are several examples:

- It appears that the XANES/EXAFS results were not measured under in-situ HER conditions. If the measurements were measured after the samples were exposed to air without reduction treatment, the Ru would be oxidized, making the comparison of Ru oxidation state meaningless. Furthermore, the EXAFs analysis was not very well done. For example, in Table S1 for VO-Ru/HfO₂-P, the coordination numbers are 11.0 for Ru-M and 4.8 for Ru-O, making the total coordination for Ru to be 15.8. That is physically impossible because the maximum coordination number of Ru is 12.

- In DFT calculations, the activation barrier for H₂O dissociation was estimated using the linear Brønsted–Evans–Polanyi (BEP) relationship from the binding energies. This method is not accurate enough for the type of comparisons the authors attempt to make in the current manuscript. The activation barriers should be calculated by properly identify the transition state over the different catalysts. For example, the authors use the estimated values of 0.65 eV for Ru/HfO₂ and 0.62 eV for VO-Ru₃/HfO₂-OP to explain that the HER activity of Ru/HfO₂ can be enhanced by introducing Vo. A difference of 0.03 eV is well within the error of calculations, in particular using the BEP approximation. Another issue with DFT is that the structural models do not seem to be consistent with the findings from EXAFS in the current manuscript.

2. This question is a general concern for many of the HER manuscripts. There have been thousands of published papers on HER. Is it worth to publish more papers on this particular topic, especially in high impact journals? The authors claim that "Importantly, Ru costs approximately half of Pt, and hence, such catalysts are economically viable". Such statement is misleading. Ru is less abundant than Pt. Therefore Ru would most likely become more expensive than Pt if Ru is used in large scale applications. Replacing

one precious metal with another precious metal, especially involving complicated synthesis procedures, would not solve any hurdles related to catalyst cost.

3. This is a relatively minor issue. The authors made extensive discussion about the shift in the XPS peak position for Ru. Similar to the concern made for XANES/EXAFS described above, Ru would be oxidized upon exposure to air, making the comparison meaningless. Furthermore, the authors appear to compare the Ru peak positions for catalysts with different particle size of Ru. The authors should check XPS literature on the "final state effect" to understand that the XPS position can also be affected by the particle size.

Reviewer #2 (Remarks to the Author):

The work represented by Meng et al. for HfO₂ supported Ru NPs for alkaline water reduction reaction is interesting and well organized. Also, all the obtained data's have been supported by relevant experimental and theoretical results. But to meet the requirement of highly respected 'Nature Communication' the MS need to be modified. I recommend 'Major revision'. My specified comments are given below.

Comments

1. During the synthesis, the as synthesized Ru/HfO₂ material was annealed under H₂/Ar atmosphere at 750°C for creating internal oxygen vacancy. Why authors particularly choose high temperature annealing method instead of simple chemical reduction method? Any specific reason?

2. The authors have used XC72 as a binder, what does it mean by XC72? Is it commercially available carbon black? Why not author can try other binder like PVDF and Nafion? Authors can compare the electrocatalytic performance by using different binders.

3. Authors can perform the electrochemical HER activity comparison of VO-Ru/HfO₂-OP materials with various oleyl amine to PVP ratio.

4. It is suggested to provide the equivalent circuit diagram for EIS outcomes.

5. All the spin orbit coupling originated peaks in XPS need to be assigned.

6. Authors saying that "owing to the larger Ru nanoparticle size of VO-Ru/HfO₂-P, which decreases the number of Ru-O-Hf bonds" in the line number of '119-120'. This decrease in number of Ru-O-Hf, needs to be supported by some experimental evidences.

7. Did authors consider the ligand field (from PVP and Oleylamine's coordination) effect while calculating the charge density profile given in Figure 4h?

8. From the theoretical studies the role of 'oxygen vacancy' is not fully understandable. Authors should give a look into it.

Reviewer #3 (Remarks to the Author):

In the manuscript "The Synergistic Effect of Hf-O-Ru Bonds and Oxygen Vacancies in Ru/HfO₂ for Enhanced Hydrogen Evolution", Li and co-authors have reported their study on the HER activity of Ru nanoparticles supported by HfO₂. They discovered that both HfO₂ and the oxygen vacancies (Vo) play a crucial role in the increase of the HER activity.

The manuscript is clear and well-written. The methodology clearly explained. I recommend the paper for publication after a few comments have been addressed.

1) The authors say that Vo play an important role for OER. What is the Vo concentration in the calculations? Is there an effect of the concentration as well?

2) Are the Vo localised on the surface of the nanocluster or deeper in the material?

3) Are the Vo participating in the adsorption of water and its splitting or they participate only by perturbing the electron distribution?

REVIEWER'S COMMENTS

Reviewer #1

In this manuscript the authors have synthesized different types of Ru/HfO₂ electrocatalysts for the hydrogen evolution reaction (HER) in alkaline electrolyte. The results suggest that the HER activity can be affected by the formation of the Ru-O-Hf bonds and by the presence of oxygen vacancies. However, there are some problems in the manuscript. The authors should address the following issues before the manuscript can be considered for publications in any journal.

1. Similar to many recent electrocatalysis papers, the authors attempt to combine the experimental HER results with XANES/EXAFS characterization and DFT calculations. This combined approach can be quite useful if it is done with expertise. However, in the current manuscript the three components are poorly integrated. Below are several examples:

- It appears that the XANES/EXAFS results were not measured under in-situ HER conditions. If the measurements were measured after the samples were exposed to air without reduction treatment, the Ru would be oxidized, making the comparison of Ru oxidation state meaningless. Furthermore, the EXAFs analysis was not very well done. For example, in Table S1 for V_O-Ru/HfO₂-P, the coordination numbers are 11.0 for Ru-M and 4.8 for Ru-O, making the total coordination for Ru to be 15.8. That is physically impossible because the maximum coordination number of Ru is 12.

Response: 1) We appreciate your insightful comments. As the reviewer remarked, in-situ XAS analysis is a powerful technique to observe the electrocatalytic behavior

during the reaction and compare the characteristics before and after the reaction. Therefore, we conducted *operando* XAS analysis under in-situ HER conditions. For *operando* XAS analysis, home-made *operando* three electrode cell system, which consist of platinum counter electrode, Hg/HgO reference electrode, and electrocatalysts loaded working electrode, with polyimide film windows were employed. The *operando* XAS results were also collected, after conducting chronoamperometry (CA) test at specific voltage in 1.0 M KOH electrolyte for 12 h.

“The Fig. 3h and i show the *operando* Ru K-edge XANES spectra and corresponding Fourier-transformed (FT) magnitudes in *operando* Ru K-edge EXAFS spectra of V_O-Ru/HfO₂-OP before and after CA testing at -0.039 V (vs. RHE) for 12 h. Evidently, both the XANES and EXAFS are similar with the initial open-circuit voltage (OCV) ones when the applied potential returned to OCV after long term CA testing, indicating the high stability.” This content has been added to page 13 of manuscript.

“In situ and *operando* XAS analysis of V_O-Ru/HfO₂-OP. In order to monitor the electronic state of the Ru active sites during the HER, potential-dependent Ru K-edge XAS measurements were performed using a home-made *operando* three electrode cell system. Fig. 4a, b and Supplementary Fig. 20 present the *operando* Ru K-edge XANES spectra recorded at different potential from open-circuit condition to -0.6 V (vs. RHE). Three important peaks labeled as pre-edge peak (1s → 4d transition), white line peak (1s → 5p transition), and maximum peak (1s → 5p transition multiple scattering), are obviously changed along with the applied potentials. The variation can be more clearly discerned from the differentiated Ru K-edge XANES intensity ($I_n^{\text{th}} - I_1^{\text{st}}$) in

Fig. 4c and d. The change of white line energy signifies the oxidation number variation, while the change of pre-edge peak and maximum peak is relevant to the degree of structural distortion. From the ex-situ sample to the OCV, a positive shift of absorption edge towards higher energy was occurred, accompanied by intensity increase of white line peak, implying the increased oxidation state of Ru³⁵. While, when cathodic potential of 0 V was applied, the absorption edge of Ru K-edge XANES spectrum was shifted to lower energy compared with the case under the OCV, along with the decreased intensity of white line, meaning a decrease of Ru oxidation state. Further to switch voltage to -0.1 V (*vs.* RHE) and -0.25 V (*vs.* RHE), the adsorption edge of Ru XANES spectra was shifted back to higher energy in relation to that under 0 V condition, and the white line intensity was also increased, demonstrating the increase of Ru oxidation state. If the more negative voltages of -0.4 V and -0.5 V were applied, the oxidation state of Ru went down again evidenced by the negative shift of adsorption edge and decreased white line intensity. Such reversible redox occurs in this way until the applied voltage back to the OCV condition. The reduction of Ru oxidation state demonstrates the electrons transfer from intermediates to Ru, which is benefit for attachment of H intermediates; While the increase of Ru oxidation state means the electrons transfer from Ru to intermediates, which is favors of the detachment of H intermediates³⁶. Thus, vigorous oxidation and reduction reactions induce the vigorous oxidation number change of Ru, making that the intermediates are easily attached and detached. As evidenced by the variations of pre-edge peaks and maximum peaks, the structural change occurred from the ex-situ sample to the OCV stage, which

corresponds to the electrode activation process. Moreover, the structural change was occurred continuously during the HER, but it is reversibly and stable. These above results indicate that the V_{O-Ru}/HfO_2-OP is flexible with respect to structural distortions and the reversible redox reaction of Ru, resulting in high catalytic activity as well as stability.

Fig. 4e and Supplementary Fig. 21 display the *operando* Ru K-edge FT-EXAFS spectra. The two main FT peaks are directly related to interatomic distances, attributing to Ru-O-Hf and Ru-Ru bonds, respectively. Apparently, the FT peak positions and intensity of Ru-O-Hf and Ru-Ru underwent a marked change during the HER catalysis. The bonds of Ru-O-Hf and Ru-Ru were contracted and stretched during reaction, and the FT peaks intensity were increased and decreased, as more clearly presented in Figs. 4f and g. Impressively, the change frequency of interatomic distances relative to Ru-O-Hf and Ru-Ru is high, however, the variation of interatomic distance changes is low, effectively demonstrating the flexible structure of V_{O-Ru}/HfO_2-OP and highly stable of it during the alkaline hydrogen electrocatalysis, which is consistent with the results of *operando* XANES spectra. Besides, the change frequency of FT peak intensity is high, but it is low for variation of intensity. This is due to the fast adsorption and desorption rate of the intermediate^{36,37}. Thus, intermediate species are easily absorbed and desorbed at Ru-O-Hf and Ru-Ru sites, bringing about fast reaction kinetics.” These analyses have been added to the pages 15-17 of manuscript. The Fig. 3 and 4 have been added to page 11 and 14, respectively, of manuscript. Supplementary Fig. 20 and 21 have been added to pages 25-26 of supporting information.

“In situ and operando XAS measurement

In situ XANES and EXAFS experiments were performed at the BL10C beam line, the Pohang Light Source (PLS-II) in Korea. A home-made *operando* three electrode cell system, which consist of platinum counter electrode, Hg/HgO reference electrode, and electrocatalysts loaded working electrode, with polyimide film windows were employed. All *operando* XAS analysis was conducted under in-situ HER conditions in 1.0 M KOH electrolyte. To detect the structural stability, the *operando* XAS results were collected after conducting chronoamperometry (CA) test at -0.039 V vs RHE for 12 h.” This in situ and *operando* XAS measurement method has been added to page 3 of supporting information.

Fig. 3 HER performance in 1.0 M KOH. **a** The polarization curves of HfO₂, V₀-Ru/HfO₂-OP, V₀-Ru/HfO₂-O, V₀-Ru/HfO₂-P, and commercial Ru/C (Ru: 5 wt%), Pt/C

(Pt: 20 wt%). **b** Tafel slopes of V_{O} -Ru/HfO₂-OP, V_{O} -Ru/HfO₂-O, V_{O} -Ru/HfO₂-P, Ru/C, and Pt/C. **c** Mass activities normalized by the noble metal mass. **d** Capacitive $\Delta j/2$ as a function of the scan rate for V_{O} -Ru/HfO₂-OP, V_{O} -Ru/HfO₂-O, V_{O} -Ru/HfO₂-P. **e** Nyquist plots for of HfO₂, V_{O} -Ru/HfO₂-OP, V_{O} -Ru/HfO₂-O, V_{O} -Ru/HfO₂-P, and Ru/C. **f** The polarization curves of Pt/C and V_{O} -Ru/HfO₂-OP before and after 5000 CV cycles. **g** The stability tests for Pt/C and V_{O} -Ru/HfO₂-OP at a constant potential of -0.039 V (vs RHE) for 100000 s. The XC-72 was used as conductive support in all measurement of HfO₂. **h** *Operando* Ru K-edge XANES spectra and **i** corresponding Fourier-transformed (FT) magnitudes in *operando* Ru K-edge EXAFS spectra of V_{O} -Ru/HfO₂-OP before and after CA testing at -0.039 V (vs. RHE) for 12 h.

Figure 4. Operando Ru K-edge XANES and EXAFS spectra of V_{O} -Ru/HfO₂-OP. a Three-dimensional plot of *operando* Ru K-edge XANES spectrum recorded at varied potential from OCV to -0.6 V (vs. RHE) during the HER catalysis. **b** The reversible change of Ru valence state during the electrocatalytic HER process: 1 refers to electrode

– OCV: oxidation; 2 refers to OCV – 0 V: reduction; 3 refers to 0 V – -0.1 V – -0.25 V: oxidation; 4 refers to -0.25 V – -0.4 V – -0.5 V: reduction; 5 refers to -0.5 V – -0.6 V: oxidation; 6 refers to -0.6 V – OCV back: reduction. **c** Three-dimensional plot and **d** Curves of normalized differentiated XANES intensity ($I_n^{\text{th}} - I_1^{\text{st}}$) in *operando* Ru K-edge XANES spectra. **e** Three-dimensional plot of *operando* Ru K-edge FT-EXAFS spectrum of V_O-Ru/HfO₂-OP. Changes in the distances and intensities of **f** Ru-Ru and **g** Ru-O-Hf in the *operando* Ru K-edge FT-EXAFS spectrum of V_O-Ru/HfO₂-OP.

Supplementary Fig. 20 a *Operando* Ru K-edge XANES spectra of V_O-Ru/HfO₂-OP, and **b** the corresponding two dimensional (2D) color map.

Supplementary Fig. 21 a *Operando* Ru K-edge FT-EXAFS spectra of V_O-Ru/HfO₂-OP, and **b** the corresponding two dimensional (2D) color map.

2) We thank the reviewer for reading the manuscript so carefully and apologize for our mistake. We found out a typing error in Supplementary Table 1. The coordination numbers of Ru-O (V_O-Ru/HfO₂-P) is 0.48. We have corrected it in revised manuscript.

Supplementary Table 1 Structure parameters extracted from the Ru K-edge EXAFS curves Fitting.

Catalysts	Bond	Coordination number (CN)	Bond length R (Å)	σ^2 (Å) x 10 ⁻³	E ₀ (eV)	R factor
Ru foil	Ru-M	12	2.662	4.0	2.5	0.00169
RuO₂	Ru-O	6	1.980	1.5	1.1	0.00099
V_O-Ru/HfO₂-O	Ru-M	5.0	2.645	8.8	4.9	0.00343
	Ru-O	2.9	2.023	1.5	2.8	0.00066
V_O-Ru/HfO₂-OP	Ru-M	4.9	2.653	8.3	4.2	0.00379
	Ru-O	3.4	2.022	1.0	2.9	0.00116
V_O-Ru/HfO₂-P	Ru-M	11.0	2.674	4.5	6.9	0.00376
	Ru-O	0.48	2.352	7.1	6.1	0.00076

Oxygen = O, Ru or Hf = M

References:

35. Cao, L. et al. Identification of single-atom active sites in carbon-based cobalt catalysts during electrocatalytic hydrogen evolution. *Nat. Catal.* **2**, 134-141 (2018).
36. Wu, Q. et al. Identifying Electrocatalytic Sites of the Nanoporous Copper–Ruthenium Alloy for Hydrogen Evolution Reaction in Alkaline Electrolyte. *ACS Energy Lett.* **5**, 192-199 (2019).
37. Chen, C. H. et al. Ruthenium-Based Single-Atom Alloy with High Electrocatalytic Activity for Hydrogen Evolution. *Adv. Energy Mater.* **9**, 1803913 (2019).

- In DFT calculations, the activation barrier for H₂O dissociation was estimated using

the linear Brønsted–Evans–Polanyi (BEP) relationship from the binding energies. This method is not accurate enough for the type of comparisons the authors attempt to make in the current manuscript. The activation barriers should be calculated by properly identify the transition state over the different catalysts. For example, the authors use the estimated values of 0.65 eV for Ru/HfO₂ and 0.62 eV for V_O-Ru₃/HfO₂-OP to explain that the HER activity of Ru/HfO₂ can be enhanced by introducing V_O. A difference of 0.03 eV is well within the error of calculations, in particular using the BEP approximation. Another issue with DFT is that the structural models do not seem to be consistent with the findings from EXAFS in the current manuscript.

Response: We very appreciate your comment. We agree with the view that the activation barriers should be calculated by properly identify the transition state over the different catalysts. To better understand the effect of oxygen vacancies on transition state over the different catalysts, HfO₂ without O defect (Ru/HfO₂), with one O defect (V_O-Ru/HfO₂-OP) and with double O defects (V_{2O}-Ru/HfO₂-OP) were calculated by properly identify the transition state according to the CI-NEB method, as shown in supplementary Figs. 33, 37, 38. The energy barrier of water dissociation for Ru/HfO₂ without O defect is 0.65 eV. For the case of V_O-Ru/HfO₂-OP, the energy barrier of water dissociation is 0.62 eV. It is even reduced to 0.54 eV for V_{2O}-Ru/HfO₂-OP, suggesting that the HER activity of Ru/HfO₂ can be enhanced by introducing O vacancy.

From the characterization of Ru/HfO₂, including HRTEM, XRD, XPS and XAS, we determine the structure of the catalyst, in which ultrasmall Ru nanoclusters combined with crystalline HfO₂. The Ru clusters with smaller size have the more

number of Ru-O-Hf bonds. The Ru cluster with 3, 6, 10, 13 atoms as simplified models were conducted, respectively, for calculation. The calculation models and XAS results are broadly in line. However, because the calculation model is simplified, it is hard to keep completely consistency with experimental data.

The relevant discussion has been added to the page 21 of manuscript and the Supplementary Fig. 38 have been added to the supporting information (page 43)

Supplementary Fig. 33 a-c Schematic energy profiles of the dissociation of H₂O by Ru/HfO₂

Supplementary Fig. 37 a-c Schematic energy profiles of the dissociation of H₂O by V₀-Ru/HfO₂-OP.

Supplementary Fig. 38 a-c Schematic energy profiles of the dissociation of H₂O by double oxygen defects system Ru/HfO₂-OP (V₂₀-Ru/HfO₂-OP).

2. This question is a general concern for many of the HER manuscripts. There have been thousands of published papers on HER. Is it worth to publish more papers on this particular topic, especially in high impact journals? The authors claim that "Importantly, Ru costs approximately half of Pt, and hence, such catalysts are economically viable". Such statement is misleading. Ru is less abundant than Pt. Therefore, Ru would most likely become more expensive than Pt if Ru is used in large scale applications. Replacing one precious metal with another precious metal, especially involving complicated synthesis procedures, would not solve any hurdles related to catalyst cost.

Response: We appreciate your comments. Based on your suggestions, we re-wrote the abstract and introduction, our original idea is whether or not Ru nanocomposite with a large band gap metal oxide, which is larger than TiO₂ (3.2 eV), can be tuned into a highly efficient HER electrocatalyst. If like this, we can distinguish the activity of Ru nanoparticles from the substrates, because some substrates such as heteroatom doped-carbon, Ni foam, they themselves have some HER activity. For these cases, it is hard to know how much activity is from Ru nanoparticles, and how much is from substrates.

The statement of “replacing platinum with Ru is because Ru is much cheaper than Pt/C” was also cancelled. We re-emphasized the advance of the work in abstract and introduction. Please see the abstract and introduction in the revised manuscript.

3. This is a relatively minor issue. The authors made extensive discussion about the shift in the XPS peak position for Ru. Similar to the concern made for XANES/EXAFS described above, Ru would be oxidized upon exposure to air, making the comparison meaningless. Furthermore, the authors appear to compare the Ru peak positions for catalysts with different particle size of Ru. The authors should check XPS literature on the "final state effect" to understand that the XPS position can also be affected by the particle size.

Response: We very appreciate your comments. We agree with the reviewer’s opinion that the Ru would be oxidized upon exposure to air, so we deleted the relevant description. Moreover, as suggested by the reviewer, we have checked the literature and reinterpreted the relationship between XPS position and particle size based on the final state effect.

“The XPS of V_O-Ru/HfO₂-OP depicts a Ru 3d_{3/2} peak, which shows a significant shift to a higher binding energy relative to that of bulk Ru (Fig. 2a). This positive core level shifts involved in the smaller metal clusters supported on less conductive substrates can be interpreted by final state effects.^{23,24} As the final state of the photoemission process, the positive hole can be less efficiently screened, leading to a positive core level shift with decreasing particle size.²⁵ Thus, the size of Ru cluster in V_O-Ru/HfO₂-OP is much smaller than that of bulk Ru. In contrast, V_O-Ru/HfO₂-P show a negative shift of 0.4 eV

compared to that of V_O-Ru/HfO₂-OP, owing to the larger Ru cluster size of V_O-Ru/HfO₂-P. The binding energy for Ru 3d_{3/2} of V_O-Ru/HfO₂-O is located in the middle of V_O-Ru/HfO₂-OP and V_O-Ru/HfO₂-P, demonstrating that the Ru cluster size in V_O-Ru/HfO₂-O is between those of V_O-Ru/HfO₂-OP and V_O-Ru/HfO₂-P. The smaller size of Ru cluster signifies more Ru-O-Hf bonds.” This content has been added to pages 6-8 of manuscript.

References:

23. Tanaka, A., Takeda, Y., Imamura, M. & Sato, S. Dynamic final-state effect on the Au 4f core-level photoemission of dodecanethiolate-passivated Au nanoparticles on graphite substrates. *Phys. Rev. B* **68**, 195415 (2003).
24. Zhang, P. & Sham, T. K. X-ray studies of the structure and electronic behavior of alkanethiolate-capped gold nanoparticles: the interplay of size and surface effects. *Phys. Rev. Lett.* **90**, 245502 (2003).
25. Lopez-Salido, I., Lim, D. C. & Kim, Y. D. Ag nanoparticles on highly ordered pyrolytic graphite (HOPG) surfaces studied using STM and XPS. *Surf. Sci.* **588**, 6-18, (2005).

Reviewer #2

The work represented by Meng et al. for HfO₂ supported Ru NPs for alkaline water reduction reaction is interesting and well organized. Also, all the obtained data's have been supported by relevant experimental and theoretical results. But to meet the requirement of highly respected 'Nature Communication' the MS need to be modified. I recommend 'Major revision'. My specified comments are given below.

Comments

1. During the synthesis, the as synthesized Ru/HfO₂ material was annealed under H₂/Ar atmosphere at 750 °C for creating internal oxygen vacancy. Why authors particularly choose high temperature annealing method instead of simple chemical reduction method? Any specific reason?

Response: We appreciate your insightful comments. There are two steps for synthesizing V_O-Ru/HfO₂-OP catalyst. First, a modified polyol process with oleylamine and polyvinylpyrrolidone as structure-directing agents was employed to prepare pristine Ru/HfO₂-OP. Second, the pristine Ru/HfO₂-OP was annealed under a H₂/Ar atmosphere, in which the volume fraction of hydrogen is 5%. In the first step, the ethylene glycol was used both as solvent and reductant, making a low oxidation state of the Ru ions in the target product and avoiding it was oxidated to a high valence state (Ru⁴⁺) (Angew. Chem. Int. Ed. 2019, 58, 13983 – 13988; J. Power Sources 2007, 168, 299–306). In the second step, the Ru/HfO₂ precursor was annealed at 750 °C under H₂/Ar (H₂, 5%) atmosphere. There are two purpose for this synthetic step: 1) creating oxygen vacancy (Adv. Mater. Interfaces 2019, 1901034; chemsuschem, 2019, 12, 2740-2747), 2) increasing the structural stability by enhancing the interaction between Ru cluster and HfO₂ substrate.

2. The authors have used XC72 as a binder, what does it mean by XC72? Is it commercially available carbon black? Why not author can try other binder like PVDF and Nafion? Authors can compare the electrocatalytic performance by using different binders.

Response: We appreciate your comments. XC 72 is a commercially available conductive carbon black (Vulcan XC 72), which can provide excellent conductivity at relatively low loading levels. Due to the high dielectric constant of HfO₂, a few XC 72 added to the catalyst system can improve the conductivity of catalyst, but the XC 72 itself has no catalytic activity. Therefore, the catalytic activity is only originated from V_O-Ru/HfO₂-OP. In addition, in our experiments, the 5 wt % Nafion solution was used as the binder.

3. Authors can perform the electrochemical HER activity comparison of V_O-Ru/HfO₂-OP materials with various oleyl amine to PVP ratio.

Response: We appreciate your comments. The effect of different ratio of O (oleylamine) to P (PVP) on the HER catalytic activity was explored. We varied the ratio of O to P from 2 : 50 to 8 : 50, and the V_O-Ru/HfO₂-OP materials prepared with the ratio of O to P of 4 : 50 exhibited the best activity with the lowest overpotential to output the benchmark current density of 10 mA cm⁻². These above results have been added to page 14 of the manuscript, and the Supplementary Fig. 19 has been added to the supporting information.

Supplementary Fig. 19 The polarization curves of the catalysts prepared with different ratio of O (oleylamine) to P (PVP).

4. It is suggested to provide the equivalent circuit diagram for EIS outcomes.

Response: We appreciate your comments. The equivalent circuit used for simulating the Nyquist plots in Fig. 3e and the corresponding electrochemical impedance parameters have been added to supporting information.

Supplementary Fig. 15 Equivalent circuit was used for simulating the Nyquist plots in Fig. 3e. R_s , R_{ct} and CPE represent the solution resistance, the charge transfer resistance and constant phase element, respectively.

Supplementary Table 2 Electrochemical impedance parameters obtained simulating the Nyquist plots to the equivalent circuit model in Supplementary Fig. 15.

Catalyst	R _s (ohm)	CPE-T (F)	CPE-P (F)	R _{ct} (ohm)
V _O -Ru/HfO ₂ -OP	9.096	0.001202	0.73969	49.1
Ru/C	9.34	0.0056874	0.47536	116.5
V _O -Ru/HfO ₂ -O	9.765	0.0021089	0.70255	200.1
V _O -Ru/HfO ₂ -P	8.017	0.0012319	0.64013	301.7
HfO ₂	10.72	0.00077802	0.79177	4276.0

5. All the spin orbit coupling originated peaks in XPS need to be assigned.

Response: We appreciate your comments. All the XPS peaks for C 1s + Ru 3d spectra have been reassigned. “The XPS of V_O-Ru/HfO₂-OP depicts a Ru 3d_{3/2} peak, which shows a significant shift to a higher binding energy relative to that of bulk Ru (Fig. 2a). This positive core level shifts involved in the smaller metal clusters supported on less conductive substrates can be interpreted by final state effects^{23,24}. As the final state of the photoemission process, the positive hole can be less efficiently screened, leading to a positive core level shift with decreasing particle size²⁵. Thus, the size of Ru cluster in V_O-Ru/HfO₂-OP is much smaller than that of bulk Ru. In contrast, V_O-Ru/HfO₂-P show a negative shift of 0.4 eV compared to that of V_O-Ru/HfO₂-OP, owing to the larger Ru cluster size of V_O-Ru/HfO₂-P. The binding energy for Ru 3d_{3/2} of V_O-Ru/HfO₂-O is located in the middle of V_O-Ru/HfO₂-OP and V_O-Ru/HfO₂-P, demonstrating that the Ru cluster size in V_O-Ru/HfO₂-O is between those of V_O-Ru/HfO₂-OP and V_O-Ru/HfO₂-P. The smaller size of Ru cluster signifies more Ru-O-Hf bonds. Besides, the three peaks around at 284.6 eV, 286.2 eV and 288.8 eV in spectra of C 1s and Ru 3d of Ru, V_O-Ru/HfO₂-O, and V_O-Ru/HfO₂-OP belong to C=C, C-O, and O-C=O, respectively,

derived from the carbon contamination on the catalysts surface²⁶. Meanwhile, the three peaks centered at 279.7 eV, 280.7 eV and 282.4 eV in the spectra of C 1s and Ru 3d of V₀-Ru/HfO₂-P are attributed to Ru 3d_{5/2} of Ru⁰, Ru⁴⁺ and Ru⁵⁺^{17,27}, respectively, indicating the possible oxidation of catalyst sample when exposed in air. While the remaining three peaks at 284.2 eV, 286.0 eV and 288.3 eV in C 1s and Ru 3d XPS spectra of V₀-Ru/HfO₂-P are assigned to C 1s originated from adsorbed carbon species.¹⁷ Those contents have been added to page 6-8 of manuscript.

References:

17. Nong, S. et al. Well-Dispersed Ruthenium in Mesoporous Crystal TiO₂ as an Advanced Electrocatalyst for Hydrogen Evolution Reaction. *J. Am. Chem. Soc.* **140**, 5719-5727 (2018).
23. Tanaka, A., Takeda, Y., Imamura, M. & Sato, S. Dynamic final-state effect on the Au4fcore-level photoemission of dodecanethiolate-passivated Au nanoparticles on graphite substrates. *Phys. Rev. B* **68**, 195415 (2003).
24. Zhang, P. & Sham, T. K. X-ray studies of the structure and electronic behavior of alkanethiolate-capped gold nanoparticles: the interplay of size and surface effects. *Phys. Rev. Lett.* **90**, 245502 (2003).
25. Lopez-Salido, I., Lim, D. C. & Kim, Y. D. Ag nanoparticles on highly ordered pyrolytic graphite (HOPG) surfaces studied using STM and XPS. *Surf. Sci.* **588**, 6-18 (2005).
26. Yu, H. et al. 2D graphdiyne loading ruthenium atoms for high efficiency water splitting. *Nano Energy* **72**, 104667 (2020).

27. Zhou, Y. et al. Lattice-confined Ru clusters with high CO tolerance and activity for the hydrogen oxidation reaction. *Nat. Catal.* **3**, 454-462 (2020).

6. Authors saying that “owing to the larger Ru nanoparticle size of V_O-Ru/HfO₂-P, which decreases the number of Ru-O-Hf bonds” in the line number of ‘119-120’. This decrease in number of Ru-O-Hf, needs to be supported by some experimental evidences.

Response: We appreciate your insightful comments. The decrease in number of Ru-O-Hf in the prepared catalysts can be attested by the advanced characterizations of Ru K-edge XANES spectra and the Fourier transforms of Ru K-edge EXAFS spectra. The fitting results of the Ru K-edge EXAFS spectra in Supplementary Table 1 show that the numbers of Ru-O-Hf bond in V_O-Ru/HfO₂-OP, V_O-Ru/HfO₂-O, and V_O-Ru/HfO₂-P are about 3.4, 2.9, 0.48, respectively, effectively demonstrating the successive decrease of Ru-O-Hf bond number in V_O-Ru/HfO₂-OP, V_O-Ru/HfO₂-O, and V_O-Ru/HfO₂-P. The weak WT signal of Ru-O-Hf for V_O-Ru/HfO₂-P also can support for the result of decrease in number of Ru-O-Hf (Fig. 2e).

Supplementary Table 1 Structure parameters extracted from the Ru K-edge EXAFS curves Fitting.

Catalysts	Bond	Coordination number (CN)	Bond length R (Å)	σ^2 (Å) x 10 ⁻³	E ₀ (eV)	R factor
Ru foil	Ru-M	12	2.662	4.0	2.5	0.00169
RuO ₂	Ru-O	6	1.980	1.5	1.1	0.00099
V _O -Ru/HfO ₂ -O	Ru-M	5.0	2.645	8.8	4.9	0.00343
	Ru-O	2.9	2.023	1.5	2.8	0.00066
V _O -Ru/HfO ₂ -OP	Ru-M	4.9	2.653	8.3	4.2	0.00379
	Ru-O	3.4	2.022	1.0	2.9	0.00116

V_O-Ru/HfO₂-P	Ru-M	11.0	2.674	4.5	6.9	0.00376
	Ru-O	0.48	2.352	7.1	6.1	0.00076

Oxygen = O, Ru or Hf = M

7. Did authors consider the ligand field (from PVP and Oleylamine's coordination) effect while calculating the charge density profile given in Figure 4h?

Response: We very appreciate your comment. We did not consider the ligand field (from PVP and Oleylamine's coordination) effect while calculating the charge density profile. The reasons are as follows: Preparation of V_O-Ru/HfO₂-OP was conducted in two continuous steps. First, a modified polyol process with oleylamine and polyvinylpyrrolidone as structure-directing agents was employed to prepare pristine Ru/HfO₂-OP. Second, the pristine Ru/HfO₂-OP was annealed at 750 °C under a H₂/Ar atmosphere for 2 h to obtain V_O-Ru/HfO₂-OP. In the first step, when the reaction was completed, the resultant products were fully washed with ethanol and cyclohexane. The aim of careful washing is to remove the ethylene glycol, PVP and Oleylamine that absorbed on the catalysts. After this step, most of the organic matter has been removed, even if there is a small amount residue, it will be pyrolyzed into carbon in the second step of high temperature calcination (750 °C for 2 h). Based on the above reasons, we think the influence of the coordination field effect could be negligible.

8. From the theoretical studies the role of 'oxygen vacancy' is not fully understandable. Authors should give a look into it.

Response: We appreciate your comment. To better understand the effect of oxygen vacancies on alkaline HER, we conduct further investigations of the influences of

oxygen vacancies. First, the oxygen vacancy localized on the surface of HfO_2 is the most stable as revealed by total energy based DFT calculation, as shown in Supplementary Fig. 25. Second, the computed adsorption energy of H_2O is -0.21 eV for $\text{HfO}_2(001)$ and -0.25 eV for $\text{V}_\text{O}\text{-HfO}_2$ (Supplementary Fig. 31), indicating that the ignorable effects of V_O on the adsorption of water. However, the computed adsorption energy of H_2O for Ru_3/HfO_2 is -0.78 eV, and the water adsorption energy of $\text{V}_\text{O}\text{-Ru}_3/\text{HfO}_2$ is -0.89 eV. These results suggest that the V_O do not directly participate in the adsorption of water but play a primary role in perturbing the electron distribution of Ru cluster. Third, the energy barrier of water dissociation for Ru/HfO_2 is 0.65 eV. For the case of $\text{V}_\text{O}\text{-Ru}/\text{HfO}_2\text{-OP}$, the energy barrier of water dissociation is 0.62 eV. It is even reduced to 0.54 eV for $\text{V}_{20}\text{-Ru}/\text{HfO}_2\text{-OP}$ (Supplementary Table 3), suggesting that the HER activity of Ru/HfO_2 can be enhanced by introducing O vacancy, and the V_O concentration also influences the HER activity. The relevant discussion has been added to the page 19, 20 and 21 of manuscript. The Supplementary Figs. 25 and 31 and Table 3 have been added to the supporting information.

Supplementary Fig. 25 Schematic of the slab models for oxygen-deficient $\text{HfO}_2(001)$ surface ($\text{Hf}_{32}\text{O}_{63}$) corresponding with the total energy.

Supplementary Fig. 31 a-e Structural representations of H_2O adsorbed on different models and the corresponding adsorption energy of H_2O .

Supplementary Table 3 Summary for the adsorption energy of Ru cluster adsorbed on

HfO₂, V_O-HfO₂ and V_{2O}-HfO₂ ($E_{\text{ads_Ru}}$); The adsorption energy of H₂O adsorbed on Ru/HfO₂, V_O-Ru₃/HfO₂-OP and V_{2O}-Ru₃/HfO₂-OP ($E_{\text{ads_H2O}}$); and the Kinetic barrier of water dissociation on the active sites of Ru/HfO₂, V_O-Ru₃/HfO₂-OP and V_{2O}-Ru₃/HfO₂-OP (E_{TS}).

	Ru/HfO₂	V_O-Ru₃/HfO₂-OP	V_{2O}-Ru₃/HfO₂-OP
E_{ads_Ru}	-5.63 eV	-7.20 eV	-8.60 eV
E_{ads_H2O}	-0.78	-0.89	-1.16 eV
E_{TS}	0.65	0.62	0.54

Reviewer #3

In the manuscript “The Synergistic Effect of Hf-O-Ru Bonds and Oxygen Vacancies in Ru/HfO₂ for Enhanced Hydrogen Evolution”, Li and co-authors have reported their study on the HER activity of Ru nanoparticles supported by HfO₂. They discovered that both HfO₂ and the oxygen vacancies (V_O) play a crucial role in the increase of the HER activity. The manuscript is clear and well-written. The methodology clearly explained. I recommend the paper for publication after a few comments have been addressed.

1) The authors say that V_O play an important role for OER. What is the V_O concentration in the calculations? Is there an effect of the concentration as well?

Response: We very appreciate your insightful comments. In the original calculation, the V_O concentration in the V_O-HfO₂ (HfO₂ with one O defect) was 1.56 %, as shown in Supplementary Fig. 26a. To better understand the effect of oxygen vacancies concentration for alkaline HER, model of HfO₂ with double O defect (V_{2O}-HfO₂), and the concentration of V_O is 3.12 % were constructed (Supplementary Fig. 27). The larger

adsorption energy of Ru₃ on V₂₀-HfO₂ (-8.60 eV) than that on V_O-HfO₂ (-7.20 eV), demonstrating that the Ru clusters supported on V₂₀-HfO₂ is more stable. The computed adsorption energy of H₂O for V_O-Ru/HfO₂ and V₂₀-Ru/HfO₂ were -0.89 and -1.16 eV, respectively, demonstrating the substantially stronger binding of water molecules to V₂₀-Ru/HfO₂ catalysts. For the case of V_O-Ru/HfO₂, the energy barrier of water dissociation is 0.62 eV (Supplementary Fig. 37). It is reduced to 0.54 eV for V₂₀-Ru/HfO₂ as shown in Supplementary Fig. 38, suggesting that the V_O concentration also significantly influences the HER catalytic activity. The above discussion has been added to the pages 19, 20, and 21 of manuscript.

Supplementary Fig. 26 **a** Structural representations of V_O-HfO₂, **b** Ru cluster adsorbed on V_O-HfO₂ and **c** H₂O adsorbed on V_O-Ru/HfO₂.

Supplementary Fig. 27 **a** Structural representations of $V_{20}\text{-HfO}_2$, **b** Ru cluster adsorbed on $V_{20}\text{-HfO}_2$ and **c** H_2O adsorbed on $V_{20}\text{-Ru/HfO}_2$.

Supplementary Fig. 37 **a-c** Schematic energy profiles of the dissociation of H_2O by $V_{20}\text{-Ru/HfO}_2\text{-OP}$.

Supplementary Fig. 38 a-c Schematic energy profiles of the dissociation of H_2O by $\text{V}_{2\text{O}}\text{-Ru/HfO}_2\text{-OP}$.

2) Are the V_O localized on the surface of the nanocluster or deeper in the material?

Response: We appreciate your comment. To make clear this question, we construct a series of models with different V_O localized position, as shown below. As confirmed by the calculated total energy, the oxygen vacancy localized on the surface of HfO_2 is the most stable. This content has been added to the page 19 of manuscript, and the Supplementary Fig. 25 was added to the supporting information.

Supplementary Fig. 25 Schematic of the slab models for oxygen-deficient $\text{HfO}_2(001)$ surface ($\text{Hf}_{32}\text{O}_{63}$) corresponding with the total energy.

3) Are the V_O participating in the adsorption of water and its splitting or they participate only by perturbing the electron distribution?

Response: We appreciate your insightful comment. In order to better understand this question, we have deeply analyzed the effect of V_O on HER process. We calculated the water adsorption energy on the HfO_2 , $\text{V}_\text{O}\text{-HfO}_2$, Ru/HfO_2 and $\text{V}_\text{O}\text{-Ru/HfO}_2$, respectively, and the results are shown in Supplementary Fig. 31. Evidently, the computed adsorption energies of H_2O on $\text{HfO}_2(001)$ and $\text{V}_\text{O}\text{-HfO}_2$ are -0.21 eV and -0.25 eV, respectively, indicating the ignorable effect of V_O on the adsorption of water. However, the computed adsorption energy of H_2O for Ru_3/HfO_2 is -0.78 eV, and it is -0.89 eV for $\text{V}_\text{O}\text{-Ru}_3/\text{HfO}_2$. These results effectively demonstrated that the V_O do not directly participate in the

adsorption of water but played a primary role in perturbing the electron distribution of Ru cluster. The relevant content has been added to the pages 20-21 of manuscript. The Supplementary Fig. 31 has been added to the supporting information.

Supplementary Fig. 31 a-e Structural representations of H₂O adsorbed on different models and the corresponding adsorption energy of H₂O.

REVIEWERS' COMMENTS

Reviewer #1 (Remarks to the Author):

The authors have addressed all my questions.

Reviewer #2 (Remarks to the Author):

The MS has been revised as requested. It can be accepted as it is now.

Reviewer #3 (Remarks to the Author):

The authors have addressed the previous comments from all the reviewers in a satisfactory way. I recommend the manuscript for publication as it is.